# Disentangled Representation Learning for Parametric Partial Differential Equations

**Ning Liu**[*]**& Lu Zhang**[*]
Department of Mathematics
Lehigh University

**Tian Gao**
IBM Research

**Yue Yu** [†]
Department of Mathematics
Lehigh University

## Abstract

Neural operators (NOs) excel at learning mappings between function spaces, serving as efficient forward solution approximators for PDE-governed systems. However, as black-box solvers, they offer limited insight into the underlying physical mechanism, due to the lack of interpretable representations of the physical parameters that drive the system. To tackle this challenge, we propose a new paradigm for learning disentangled representations from NO parameters, thereby effectively solving an inverse problem. Specifically, we introduce DisentangO, a novel hyper-neural operator architecture designed to unveil and disentangle latent physical factors of variation embedded within the black-box neural operator parameters. At the core of DisentangO is a multi-task NO architecture that distills the varying parameters of the governing PDE through a task-wise adaptive layer, alongside a variational autoencoder that disentangles these variations into identifiable latent factors. By learning these disentangled representations, DisentangO not only enhances physical interpretability but also enables more robust generalization across diverse systems. Empirical evaluations across supervised, semi-supervised, and unsupervised learning contexts show that DisentangO effectively extracts meaningful and interpretable latent features, bridging the gap between predictive performance and physical understanding in neural operator frameworks. Our code and data accompanying this paper are available at https://github.com/ningliu-iga/DisentangO.

## 1 Introduction

Interpretability in machine learning (ML) refers to the ability to understand and explain how models make decisions (Rudin et al., 2022; Molnar, 2020). As ML systems grow complex, especially with the use of deep learning and ensemble methods (Sagi & Rokach, 2018), the reasoning behind their predictions can become opaque. Interpretability addresses this challenge by making the models more transparent, enabling users to trust the outcomes, detect biases, and identify potential flaws. It is a critical factor in applying AI responsibly, especially in fields where accountability and fairness are essential (Cooper et al., 2022). In physics, where models endeavor to capture the governing laws and physical principles, understanding how a model arrives at its predictions is vital for verifying that it aligns with known scientific theories. This transparency is key for advancing scientific discovery, validating results, and enhancing trust in models for complex physical systems.

Discovering interpretable representations of physical parameters in learning physical systems is challenging due to the intricate nature of real-world phenomena and the often implicit relationships between variables. In physics, quantities like force, energy, and velocity are governed by well-established laws, and extracting them in a way that aligns with physical intuition requires models that go beyond mere pattern recognition. Traditional ML models may fit the data but fail to provide a physically interpretable way. To address this, recent developments include integrating physical constraints into the learning process (Raissi et al., 2019), such as embedding conservation laws (Liu et al., 2023; 2024a), symmetries (Mattheakis et al., 2019), or invariances (Cohen & Welling, 2016) directly into model architectures. Additionally, methods like symbolic regression (Biggio et al.,

---

[*]Equal contribution
[†]Corresponding author, yuy214@lehigh.edu

2021) and sparse modeling (Carroll et al., 2009) aim to discover simple, interpretable expressions that capture the underlying dynamics. However, balancing model expressivity with interpretability remains a significant challenge, as overly complex models may obscure the true physical relationships, while oversimplified models risk losing critical details of the system's behavior.

We introduce **DisentangO**, a novel variational hyper-neural operator architecture to disentangle physical factors of variation from black-box neural operator parameters for solving parametric PDEs. Neural operators (NOs) (Li et al., 2020a;c) learn mappings between infinite-dimensional function spaces in the form of integral operators, making them powerful tools for discovering continuum physical laws by manifesting the mappings between spatial and/or spatiotemporal data; see You et al. (2022a); Liu et al. (2024a;b; 2023); Ong et al. (2022); Cao (2021); Lu et al. (2019; 2021); Goswami et al. (2022); Gupta et al. (2021) and references therein. However, most NOs serve as efficient forward surrogates for the underlying physical system under a supervised learning setting. As a result, they act as black-box universal approximators for a single physical system governed by a fixed set of PDE parameters, and lack interpretability with respect to the underlying physical laws. In contrast, the key innovation of DisentangO lies in the use of a hypernetwork architecture that distills the varying physical parameters of the governing PDE from multiple physical systems through an unsupervised learning setting. The distilled variables are further disentangled into distinct physical factors to enhance physical understanding and promote robust generalization. Consequently, DisentangO effectively extracts meaningful and interpretable physical features, thereby simultaneously solving both the forward and inverse problems. **Our key contributions** are:

- We bridge the divide between predictive accuracy and physical interpretability, and introduce a new paradigm that simultaneously performs physics modeling (i.e., as a forward PDE solver) and governing physical mechanism discovery (i.e., as an inverse PDE solver).

- We propose a novel variational hyper-neural operator architecture, which we coin DisentangO. DisentangO extracts the key physical factors of variation from black-box neural operator parameters of multiple physical systems. These factors are then disentangled into distinct latent factors that enhance physical interpretation and generalization across different physical systems.

- We provide theoretical analysis on the component-wise identifiability of the true generative factors in physics modeling: by learning from multiple physical systems, the variability of hidden physical states in these systems promotes identifiability.

- We explore the practical utility of disentanglement and perform experiments across a broad range of settings including supervised, semi-supervised, and unsupervised learning. Results show that DisentangO effectively extracts meaningful physical features.

## 2 BACKGROUND AND RELATED WORK

**Neural operators.** Learning complex physical systems from data is essential in many scientific applications (Carleo et al., 2019; Liu et al., 2024c; Karniadakis et al., 2021; Zhang et al., 2018; Cai et al., 2022; Pfau et al., 2020; He et al., 2021; Jafarzadeh et al., 2024). When governing laws are unknown, models must be both *resolution-invariant* for consistent performance across discretizations and *interpretable* for domain experts. Neural operators (NOs) achieve the former by learning mappings between infinite-dimensional function spaces (Li et al., 2020a;b;c; You et al., 2022a; Ong et al., 2022; Cao, 2021; Lu et al., 2019; 2021; Goswami et al., 2022; Gupta et al., 2021), enabling accurate and consistent predictions of continuum physical surrogates. However, NOs lack interpretable representations of physical states, limiting their ability to reveal underlying physical mechanisms.

**Hypernetworks.** Hypernetworks (Ha et al., 2016; Chauhan et al., 2024) are a class of neural network (NN) architectures that use one NN to generate weights for another NN, both trained in an end-to-end manner. This allows soft weight sharing across tasks, benefiting transfer learning and dynamic information sharing (Chauhan et al., 2023). Hypernetworks can also be employed as a versatile technique in existing NN architectures. For instance, in Nguyen et al.; Oh & Peng (2022), a hypernetwork is employed to generate parameters for a VAE model and enable multi-task learning. Similarly, Lee et al. (2023) integrates the hypernetwork architecture with neural operators. However, none of the existing work discusses the capability of hypernetworks in hidden physics discovery.

**Forward and inverse PDE learning.** Existing NOs serve as efficient surrogates for forward PDE solving but often act as black-box approximators, lacking interpretability. In contrast, deep learning methods for inverse PDE solving (Fan & Ying, 2023; Molinaro et al., 2023; Jiang et al., 2022;

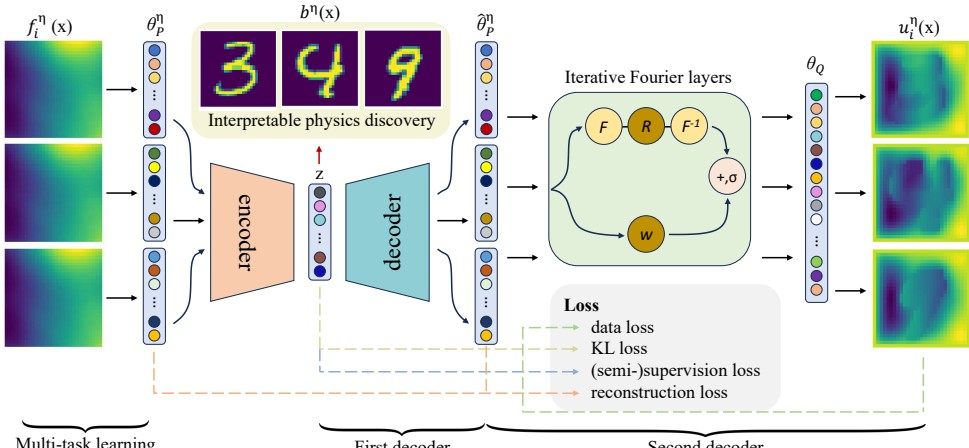

Figure 1: Overview of the DisentangO architecture. Each task correspond to a different (hidden) PDE parameter $\boldsymbol{b}$. For illustration, the same input function $f_i^\eta$ is shown for multiple tasks to highlight that different parameter fields $b^\eta$ can produce different output functions $u_i^\eta$ under identical input $f$; in practice, $f$ may vary across tasks. The task-specific lifting parameters $\theta_P^\eta$ are encoded and reconstructed through a VAE, and the reconstructed parameters $\hat{\theta}_P^\eta$ are fed into the iterative Fourier layers to form the task-specific neural operator $G^\eta$. Loss components are overlaid to indicate where each term in the objective $L_{loss}$ is computed.

Chen et al., 2023) aim to reconstruct PDE parameters from solution data but face challenges due to ill-posedness. To address this, many NOs incorporate prior information via governing PDEs (Yang et al., 2021; Li et al., 2021), regularizers (Dittmer et al., 2020; Obmann et al., 2020; Ding et al., 2022; Chen et al., 2023), or structured operators (Lai et al., 2019; Yilmaz, 2001). However, these methods assume prior knowledge of the model form, which is often unrealistic. A recent approach (Yu et al., 2024) employs attention to construct a data-dependent kernel for inverse mapping but does not disentangle the learned kernel or extract interpretable parameters. To our knowledge, DisentangO is the first to tackle both forward (physics prediction) and inverse (physics discovery) PDE learning while simultaneously identifying distinct physical parameters from the learned NO.

**Disentangled representation learning.** Disentangled representation learning separates data into distinct, interpretable factors, each capturing an independent variation. It has critical implications for transfer learning, generative modeling, and AI fairness, enabling models to generalize by leveraging isolated features. Key advances have been driven by models like $\beta$-VAE (Higgins et al., 2017), FactorVAE (Kim & Mnih, 2018), and InfoGAN (Chen et al., 2016), which promote disentanglement through latent regularization and mutual information constraints. While early work focused on supervised or semi-supervised methods (Ridgeway & Mozer, 2018; Shu et al., 2019; Mathieu et al., 2019), recent efforts target unsupervised learning (Duan et al., 2019), though challenges remain (Locatello et al., 2019), with ongoing efforts to improve metrics, robustness, and applicability to complex data. Most research has centered on computer vision and robotics, where latent factors have human-interpretable visual meanings, while its exploration in physical system learning remains limited (Lingsch et al., 2024; Tong et al., 2024; Fotiadis et al., 2023). Moreover, existing work disentangles from data, whereas our work is the first to disentangle from black-box NN parameters.

## 3 DISENTANGO

We consider a series of complex systems with different hidden physical parameters:

$$\mathcal{K}_{\boldsymbol{b}}[\boldsymbol{u}](\boldsymbol{x}) = \boldsymbol{f}(\boldsymbol{x}), \quad \boldsymbol{x} \in \Omega . \tag{3.1}$$

Here, $\Omega \subset \mathbb{R}^s$ is the domain of interest, $\boldsymbol{f}(\boldsymbol{x})$ is a function representing the loading on $\Omega$, $\boldsymbol{u}(\boldsymbol{x})$ is the corresponding solution of this system. $\mathcal{K}_{\boldsymbol{b}}$ represents the unknown governing law, e.g., balance laws, determined by the (possibly unknown and high-dimensional) parameter field $\boldsymbol{b}$. For instance, in a material modeling problem, $\mathcal{K}_{\boldsymbol{b}}$ often stands for the constitutive law and $\boldsymbol{b}$ can be a vector ($\boldsymbol{b} \in \mathbb{R}^{d_b}$) representing the homogenized material parameter field or a vector-valued function ($\boldsymbol{b} \in L^\infty(\Omega; \mathbb{R}^{d_b})$) representing the heterogeneous material properties. Both scenarios are considered in our empirical experiments in Section 4.

Many physical modeling tasks can be formulated as either a forward or an inverse PDE-solving problem. In a forward problem setting, one aims to find the PDE solution when given PDE informa-

tion, including coefficient functions, boundary conditions, initial conditions, and loading sources. That means, given the governing operators $\mathcal{K}$, the parameter (field) $\boldsymbol{b}$, and loading field $\boldsymbol{f}(\boldsymbol{x})$ in equation 3.1, the goal is to solve for the corresponding solution field $\boldsymbol{u}(\boldsymbol{x})$, through classical PDE solvers (Brenner & Scott, 2007) or data-driven approaches (Lu et al., 2019; Li et al., 2020c). As a result, a forward map is constructed:

$$\mathcal{G} : (\boldsymbol{b}, \boldsymbol{f}) \to \boldsymbol{u} . \tag{3.2}$$

Here, $\boldsymbol{b}$ and $\boldsymbol{f}$ are input vectors/functions, and $\boldsymbol{u}$ is the output function.

Conversely, solving an inverse PDE problem involves reconstructing the underlying full or partial PDE information from PDE solutions, where one seeks to construct an inverse map:

$$\mathcal{H} : (\boldsymbol{u}, \boldsymbol{f}) \to \boldsymbol{b} . \tag{3.3}$$

Unfortunately, solving an inverse problem is typically more challenging due to the ill-posed nature of the PDE model. In general, a small number of function pairs $(\boldsymbol{u}, \boldsymbol{f})$ from a single system does not suffice in inferring the underlying parameter field $\boldsymbol{b}$, making the inverse problem generally non-identifiable (Molinaro et al., 2023).

Herein, we propose a novel neural architecture to alleviate the curse of ill-posedness without access to the exact governing partial differential equation (PDE). The key ingredients are: 1) the construction of a multi-task NO architecture, which solves both the forward (physics prediction) problems simultaneously from multiple PDE systems with different hidden parameters; 2) a generative model to disentangle the key features from NO parameters that contain critical information of $\boldsymbol{b}$, as the inverse (physics discovery) problem solver.

## 3.1 Variational hyper-neural operator as a multi-task solver

**Notation and data model assumptions**. We denote by $\boldsymbol{f}(\boldsymbol{x})$ and $\boldsymbol{u}(\boldsymbol{x})$ the input and output functions of the NO, respectively. Let $p(\cdot)$ denote a probability density function, $\mathbb{E}(\cdot)$ the expectation, and $||\cdot||$ the $l^2$-norm. We assume (noisy) observations of both $\boldsymbol{u}$ and $\boldsymbol{f}$ are available on a common physical domain $\Omega$, where $\boldsymbol{u} \in \mathcal{U} \subset L^2(\Omega; \mathbb{R}^{d_u})$ and $\boldsymbol{f} \in \mathcal{F} \subset L^2(\Omega; \mathbb{R}^{d_f})$. Here, $\mathcal{U}$ and $\mathcal{F}$ are the Banach spaces of solution fields and loading fields, respectively. Formally, we consider $S$ training datasets $\mathcal{D}^\eta$, $\eta = 1, \dots, S$, each corresponding to the same PDE equation 3.1 but with a different (hidden) physical parameter $\boldsymbol{b}^\eta \in \mathcal{B}$, with $\mathcal{B}$ the parameter space. In our multi-task learning model, each dataset corresponds to one task. We assume the hidden parameter is generated according to:

$$\boldsymbol{b} \sim \mathbb{P}_b , \, \boldsymbol{z} \sim p(\boldsymbol{z}|\boldsymbol{b}) , \tag{3.4}$$

where $\boldsymbol{z} \in \mathcal{Z} \subset \mathbb{R}^{d_z}$ is the latent embedding of $\boldsymbol{b}$. Each dataset contains measurements from $n_{train}^\eta$ function pairs $\{(\boldsymbol{u}_i^\eta(\boldsymbol{x}), \boldsymbol{f}_i^\eta(\boldsymbol{x}))\}_{i=1}^{n_{train}^\eta}$. Although our method accommodates datasets with varying numbers of function pairs, we use the same number of function pairs in this work for simplicity and denote $n_{train}^\eta = n_{train}$. While actual observations are gathered on a discrete sensor set $\chi = \{\boldsymbol{x}_j\}_{j=1}^{\#\chi}$ and inevitably contain observational noise of the solution, we adopt the standard assumption that this noise is additive and i.i.d. Letting $\mathcal{G}^\dagger[\boldsymbol{f}; \boldsymbol{b}]$ denote the true forward operator of equation 3.1, we write the data model associated with multi-task learning as:

$$\mathcal{D}^\eta = \{\{(\boldsymbol{u}_{i,j}^\eta, \boldsymbol{f}_{i,j}^\eta)\}_{j=1}^{\#\chi}\}_{i=1}^{n_{train}} , \mathcal{D} = \bigcup_\eta \mathcal{D}^\eta \tag{3.5}$$

with

$$\boldsymbol{f} \sim \mathbb{P}_f, \, \boldsymbol{f}_{i,j}^\eta = \boldsymbol{f}_i^\eta(\boldsymbol{x}_j), \, \boldsymbol{u}_{i,j}^\eta = \boldsymbol{u}_i^\eta(\boldsymbol{x}_j) + \epsilon_{\eta,i,j} = \mathcal{G}^\dagger[\boldsymbol{f}_i^\eta; \boldsymbol{b}^\eta](\boldsymbol{x}_j) + \epsilon_{\eta,i,j}, \, \epsilon_{\eta,i,j} \sim \mathcal{N}(0, \varpi^2). \tag{3.6}$$

The forward modeling objective is to learn a surrogate solution operator $\mathcal{G}[\cdot; \theta^\eta]$ for $\mathcal{G}^\dagger$, where $\theta^\eta$ denotes the task-specific NO parameters for the forward surrogate operator in the $\eta$-th task, associated with $\boldsymbol{b}^\eta$. Note that all tasks share the same NO architecture, $\mathcal{G}$, and the surrogate operator for each task depends on the physical system $\eta$ through the task-specific parameter $\theta^\eta$.

For inverse modeling, let $\mathcal{H}^\dagger$ denote the (unknown) inverse operator that maps function pairs $(\boldsymbol{u}, \boldsymbol{f})$ to the underlying PDE parameter $\boldsymbol{b}$. In practice, identifying $\mathcal{G}^\dagger$ is often impossible, as the available data of $(\boldsymbol{u}, \boldsymbol{f})$ may not contain sufficient information to identify all features of $\boldsymbol{b}$. For instance, in a Dirichlet boundary condition problem, $\boldsymbol{u}_i^\eta(\boldsymbol{x}) = 0$ for all $\boldsymbol{x} \in \partial\Omega$, making $\boldsymbol{b}^\eta$ not learnable on $\partial\Omega$. A more realistic goal is therefore to recover the underlying mechanism of $\boldsymbol{b}$ in the space of identifiability, i.e., $\boldsymbol{z}$. Thus, our second objective is to construct an inverse map by estimating embedding $\tilde{\boldsymbol{z}}^\eta$ of the hidden parameter $\boldsymbol{b}^\eta$ from $\theta^\eta$:

$$\mathcal{H}(\theta^\eta; \Theta_H) \approx \tilde{\boldsymbol{z}}^\eta , \tag{3.7}$$

where $\Theta_H$ are trainable parameters of $\mathcal{H}$, and $\tilde{z}^\eta \in \mathbb{R}^{d_z}$ denotes the learned latent variables that can be transformed to the ground-truth latent variable $z^\eta$ via an invertible function $h$. The goal of disentanglement is to discover $\tilde{z}$, together with the solution operator $\mathcal{G}$.

The first objective learns the forward operator $\mathcal{G}[\cdot; \theta^\eta]$ in a supervised fashion, in the form of function-to-function mappings, for all (hidden) parameters $b^\eta$ in the range of interest; while the second objective learns the vector-to-vector mapping from $\theta^\eta$ to $\tilde{z}^\eta$, which is unsupervised due to the hidden physics nature. We propose to employ a variational autoencoder (VAE) (Kingma & Welling, 2013) as the representation learning approach for the second objective, and a meta-learned NO architecture as a universal solution operator for the first objective*. The key is to pair these architectures as a hyper-neural operator, which simultaneously solves both forward and inverse problems. Although the proposed strategy is generic and thus applicable to other multi-task neural operator architectures, to provide a universal architecture of $\mathcal{G}$ for different tasks, we adopt a meta-learning strategy following the meta-learned neural operator (MetaNO) (Zhang et al., 2023). MetaNO is developed based on the implicit Fourier neural operator (IFNO), a PDE solution operator with a relatively small number of trainable parameters (You et al., 2022b). In MetaNO, task-wise adaptation is applied only to the trainable parameters in the first layer of the NO, whereas all other parameters are shared across tasks. For an $L$-layer MetaNO, we write:

$$\mathcal{G}[f; \theta^\eta](x) = \mathcal{G}[f; \theta_P^\eta, \theta_J, \theta_Q](x) := \mathcal{Q}_{\theta_Q} \circ (\mathcal{J}_{\theta_J})^L \circ \mathcal{P}_{\theta_P^\eta}[f](x) , \qquad (3.8)$$

where $\mathcal{P}, \mathcal{Q}$ are shallow-layer MLPs that map a low-dimensional vector to a high-dimensional vector and vice versa, parameterized by $\theta_P^\eta$ and $\theta_Q$, respectively. Each intermediate layer, $\mathcal{J}$, is constructed as a mimetic of a fixed-point iteration step and parameterized by $\theta_J$. Supported by the universal approximator analysis (Zhang et al., 2023), different PDEs share common iterative ($\theta_J$) and projection ($\theta_Q$) parameters, with all the information about parameter $b$ encoded in the task-wise lifting parameters $\theta_P^\eta$. Hence, to provide an inverse map from the task-wise NO parameter $\theta^\eta$ to the key features of the PDE parameters, $\tilde{z}^\eta$, one can construct $\mathcal{H}$ as the mapping from $\theta_P^\eta$ to $\tilde{z}^\eta$ using a neural network architecture. In this work, we employ a standard MLP:

$$\mathcal{H}(\theta^\eta; \Theta) := \text{MLP}(\theta_P^\eta) , \qquad (3.9)$$

since all other NO parameters are invariant to the change of $b$. This construction substantially reduces the degrees of freedom in $\theta^\eta$, making the invertibility assumption in the next section feasible. To simplify notation, we use $\theta^\eta$ to denote $\theta_P^\eta$ in the subsequent discussion. An overview of the forward and inverse operator architecture is provided in Figure 1.

We now define the learning objective. The overall objective is to maximize the log data likelihood:

$$\max \mathbb{E}(\log p(\theta, \mathcal{D})) = \max \left[ \mathbb{E}\big( \log p(u|f, \theta) + \log p(\theta|f) + \log p(f) \big) \right].$$

Note that $p(f)$ remains constant over different NO parameter $\theta$, and the assumption in eq. 3.6 yields $\log(p(\theta|f)) = \log(p(\theta))$. Additionally, the assumption in eq. 3.4 guarantees that $\theta$ is generated from the latent space over $z$, hence $\log(p(\theta)) = \log \int_z p(\theta|z)p(z)dz$. The overall objective then becomes:

$$\max \mathbb{E}(\log(p(\theta, \mathcal{D}))) = \max \left[ \mathbb{E}(\log(p(u|f, \theta))) + \mathbb{E}\big( \log \int_z p(\theta|z)p(z)dz \big) \right]. \qquad (3.10)$$

However, the second term in this formulation is generally intractable. We use a variational posterior $q(\theta|z)$ to approximate the actual posterior $p(\theta|z)$ and maximize the evidence lower bound (ELBO):

$$L_{ELBO} = \frac{1}{S} \sum_{\eta=1}^{S} \left[ \mathbb{E}_{q(z^\eta|\theta^\eta)} \log p(\theta^\eta|z^\eta) - D_{KL}\left( q(z^\eta|\theta^\eta) || p(z^\eta) \right) \right] ,$$

with $D_{KL}(\cdot||\cdot)$ denoting the KL divergence between two distributions. Putting everything together, we obtain the loss functional:

$$L_{loss} = \frac{1}{S} \sum_{\eta=1}^{S} \left[ -\mathbb{E}(\log(p(u|f, \theta^\eta))) - \mathbb{E}_{q(z^\eta|\theta^\eta)} \log p(\theta^\eta|z^\eta) + D_{KL}\left( q(z^\eta|\theta^\eta) || p(z^\eta) \right) \right].$$
$$(3.11)$$

---

*Note that tasks differ in the PDE coefficients (the physical parameters $b^\eta$) and correspondingly the NO parameter $\theta^\eta$, not in the NO architecture or its input-output structure.

The above formulation naturally lends itself to a hierarchical variational autoencoder (HVAE) architecture (Vahdat & Kautz, 2020). The encoder aims to obtain the posterior $q_{\mu_z, \Sigma_z}(\boldsymbol{z}^\eta | \theta^\eta)$ and provide the inverse map $\mathcal{H}$:

$$\hat{\boldsymbol{z}} \sim q_{\mu_z, \Sigma_z}(\hat{\boldsymbol{z}}^\eta | \theta^\eta) \,. \tag{3.12}$$

Note that although $\mathcal{H}$ represents a deterministic inverse mapping from the NO parameters to the latent variables, in equation 3.12 we approximate this mapping using a probabilistic encoder to account for uncertainty and enable variational inference. This follows standard practice in the VAE literature, where deterministic relationships are modeled probabilistically for tractable inference.

Then, the first decoder $\hat{g}$ processes the estimated latent variable $\hat{\boldsymbol{z}}$ and reconstructs the corresponding NO parameter $\hat{\theta}$:

$$\hat{\theta} = \hat{g}(\hat{\boldsymbol{z}}) \,. \tag{3.13}$$

Lastly, the second decoder reconstructs the forward map $\mathcal{G}$, by further taking the estimated $\hat{\theta}$ and the loading function $\boldsymbol{f}$ and estimating the output function $\boldsymbol{u}$:

$$\hat{\boldsymbol{u}} = \hat{\mathcal{G}}[\boldsymbol{f}; \hat{\theta}] \,. \tag{3.14}$$

## 3.2 DISENTANGLING THE UNDERLYING MECHANISM

DisentangO aims to identify and disentangle the components of the latent representation $\boldsymbol{z}$, which serves as an inverse PDE solver. However, to capture the true physical mechanism, a natural question arises: is it really possible to identify the latent variables of interest (i.e., $\boldsymbol{z}$) with only observational data $\{\mathcal{D}^\eta\}_{\eta=1}^S$? We now show that, by learning the model $(p_{\hat{\boldsymbol{z}}}, \hat{g}, \hat{\mathcal{H}}, \hat{\mathcal{G}})$ that matches the true marginal data distribution in all domains, we can indeed achieve this identifiability for the generating process proposed in equation 3.4 and equation 3.6. Before the formal theorem, we first state our assumptions:

**Assumption 1.** (Density Smoothness and Positivity) The probability density functions for $\theta$ and $\boldsymbol{z}$, which we denote as $p_\theta$ and $p_{\boldsymbol{z}}$, are both smooth and positive.

**Assumption 2.** (Invertibility) The task-wise parameter $\theta$ can be generated by $\boldsymbol{z}$ through an invertible and smooth function $\mathcal{H}^{-1}$. Moreover, for each given $\boldsymbol{f}$, we denote $\mathcal{G}_f(\theta) := \mathcal{G}[\boldsymbol{f}; \theta](\boldsymbol{x})$ as the operator mapping from $\mathbb{R}^{d_\theta}$ to $\mathcal{U}$. $\mathcal{G}_f$ is also one-to-one with respect to $\theta$.

**Assumption 3.** (Conditional Independence) Conditioned on $\boldsymbol{b}$, each component of $\boldsymbol{z}$ is independent of each other: $\log p_{\boldsymbol{z}|\boldsymbol{b}}(\boldsymbol{z}|\boldsymbol{b}) = \sum_{i=1}^{d_z} \log p_{z_i|\boldsymbol{b}}(z_i|\boldsymbol{b})$.

**Assumption 4.** (Linear independence) There exists $2d_z + 1$ values of $\boldsymbol{b}$, such that the $2d_z$ vectors $w(\boldsymbol{z}, \boldsymbol{b}^j) - w(\boldsymbol{z}, \boldsymbol{b}^0)$ with $j = 1, \cdots, 2d_z$ are linearly independent. Here,

$$w(\boldsymbol{z}, \boldsymbol{b}^j) := \left( \frac{\partial q_1(z_1, \boldsymbol{b}^j)}{\partial z_1}, \cdots, \frac{\partial q_{d_z}(z_{d_z}, \boldsymbol{b}^j)}{\partial z_{d_z}}, \frac{\partial^2 q_1(z_1, \boldsymbol{b}^j)}{\partial z_1^2}, \cdots, \frac{\partial^2 q_{d_z}(z_{d_z}, \boldsymbol{b}^j)}{\partial z_{d_z}^2} \right), \tag{3.15}$$

with $q_i(z_i, \boldsymbol{b}) := \log p_{z_i|\boldsymbol{b}}$.

First, we show that the latent variable $\boldsymbol{z}$ can be identified up to an invertible component-wise transformation: for the true latent variable $\boldsymbol{z} \in \mathbb{R}^{d_z}$, there exists an invertible function $h : \mathbb{R}^{d_z} \to \mathbb{R}^{d_z}$, such that $\hat{\boldsymbol{z}} = h(\boldsymbol{z})$.

**Theorem 1.** We follow the data-generating process in equation 3.4 and equation 3.6, and Assumptions 1-2. Then, by learning $(p_{\hat{\boldsymbol{z}}}, \hat{g}, \hat{\mathcal{H}}, \hat{\mathcal{G}})$ to achieve:

$$p_{\hat{\boldsymbol{u}}|\boldsymbol{f}} = p_{\boldsymbol{u}|\boldsymbol{f}} \,, \tag{3.16}$$

where $\boldsymbol{u}$ and $\hat{\boldsymbol{u}}$ are generated from the true process and the estimated model, respectively, $\boldsymbol{z}$ is identifiable up to an invertible function $h$.

**Proof:** Please see Appendix A.

Additionally, with additional assumptions on conditional independence and datum variability, we can further obtain the following theoretical results on component-wise identifiability: for each true component $z_i$, there exists a corresponding estimated component $\hat{z}_j$ and an invertible function $h_i : \mathbb{R} \to \mathbb{R}$, such that $z_i = h_i(\hat{z}_j)$.

**Theorem 2.** We follow the data-generating process in equation 3.4 and equation 3.6 and Assumptions 1-4. Then, by learning $(p_{\hat{\boldsymbol{z}}}, \hat{g}, \hat{\mathcal{H}}, \hat{\mathcal{G}})$ to achieve equation 3.16, $\boldsymbol{z}$ is component-wise identifiable.

**Proof:** Please see Appendix A.

**Discussion on assumptions.** Intuitively, Assumptions 1-2 are required to guarantee that there exists a smooth and injective mapping between the ground-truth latent embedding $z$ and the learned embedding $\hat{z}$. Assumption 4 indicates sufficient variability across physical systems. This is a common assumption in the nonlinear ICA literature for domain adaptation (Hyvarinen et al., 2019; Khemakhem et al., 2020; Kong et al., 2023). Further discussions and validation on the assumptions are provided in Appendix A. To our best knowledge, it is the first time the component-wise identifiability is discussed in the context of multi-task neural operator learning.

### 3.3 A GENERIC ALGORITHM FOR VARIOUS SUPERVISION CASES

Although DisentangO is primarily designed for the challenging scenario of learning without supervision on $b$ nor prior knowledge on the PDE model form in equation 3.1, its methodology is generic and readily can be extended to handle the scenarios with partial or full measurements of $b$, which are common in classical inverse PDE benchmark problems. In this section, we discuss the practical utility of DisentangO under three scenarios:

- (SC1: Supervised) The value of $b^\eta$ is given.
- (SC2: Semi-supervised) The value of $b^\eta$ is not given, but a label $c(b^\eta)$ (e.g., a classification of $b^\eta$) is given.
- (SC3: Unsupervised) No value or label is given for $b^\eta$ for each task.

A pseudo Algorithm 1 is provided in the Appendix.

To obtain the posterior $q$ in equation 3.12, we assume that each latent variable satisfies a Gaussian distribution of distinct means and diagonal covariance, i.e., $q_{\mu_z, \Sigma_z}(\hat{z}|\theta) := \mathcal{N}(\mu_z(\theta), \Sigma_z^2(\theta))$, then estimate its mean and covariance using an MLP. As a result, the KL-divergence term in the ELBO admits a closed form:

$$\text{SC2/SC3:} \quad D_{KL}\left(q(\hat{z}|\theta) \parallel p(z)\right) = \frac{1}{2} \sum_{i=1}^{d_z} \left((\Sigma_z)_i^2 + (\mu_z)_i^2 - 2\log((\Sigma_z)_i) - 1\right). \quad (3.17)$$

In the supervised setting, we take $\mu_z(\theta_P^\eta) = b^\eta$, and then the KL divergence term writes:

$$\text{SC1:} \quad D_{KL}\left(q(\hat{z}|\theta) \parallel p(z)\right) = \frac{1}{2} \sum_{i=1}^{d_z} \left((\Sigma_z)_i^2 + (\mu_z - b)_i^2 - 2\log((\Sigma_z)_i) - 1\right). \quad (3.18)$$

For the first decoder, the likelihood $p(\theta^\eta|z^\eta)$ is a factorized Gaussian with mean $\mu_\theta$ and covariance $\Sigma_\theta$, computed from another MLP. By taking as input Monte Carlo samples once for each $z$, the reconstruction accuracy term can be approximated as:

$$E_{q(z|\theta)} \log p(\theta|z) \approx -\sum_{i=1}^{d_\theta} \left(\frac{((\theta)_i - (\mu_\theta)_i)^2}{2(\Sigma_\theta)_i^2}\right) - \sum_{i=1}^{d_\theta} (\log(\Sigma_\theta)_i + c) \ ,$$

where $d_\theta$ is the dimension of $\theta$, and $c$ is a constant.

For the second decoder, we parameterize it as a multi-task NO in equation 3.8, and model the discrepancy between $\hat{\mathcal{G}}[f_j^\eta; \hat{\theta}^\eta](x_k)$ and the ground truth $u(x_k)$ as an additive independent unbiased Gaussian random noise $\epsilon$, with

$$\hat{\mathcal{G}}[f_j^\eta; \hat{\theta}^\eta](x_k) = u_j^\eta(x_k) + \epsilon_{\eta,j,k}, \epsilon_{\eta,j,k} \sim \mathcal{N}(0, \varpi^2),$$

where $\varpi$ is the standard deviation of the additive noise as defined in the data model. In practice, the observational noise is unknown, and we treat $\varpi$ as a tunable hyperparameter. Then, with a uniform spatial discretization of size $\Delta x$ in a domain $\Omega \subset \mathbb{R}^{d_u}$, the log likelihood after eliminating the constant terms can be written as:

$$\frac{1}{2\varpi^2 \Delta x^{2d_u}} \sum_{\eta=1}^{S} \sum_{j=1}^{n_{train}} \left\|\hat{\mathcal{G}}[f_j^\eta; \hat{\theta}^\eta](x_k) - u_j^\eta(x_k)\right\|_{L^2(\Omega)}^2. \quad (3.19)$$

In our empirical tests in Section 4, for the unsupervised and semi-supervised scenarios, we choose $q$ as a standard Gaussian distribution following the independence assumption of Theorem 2. Additionally, to avoid overparameterization, we take $\Sigma_\theta = \sigma_\theta^2 \mathbf{I}$, with $\sigma_\theta$ being a tunable hyperparameter.

In this case, the total objective can be further simplified as:

$$L_{loss} = \frac{1}{S} \sum_{\eta=1}^{S} \left( \beta_d \sum_{j=1}^{n_{train}} \left\| \hat{\mathcal{G}}[\boldsymbol{f}_j^\eta; \hat{\theta}^\eta] - \boldsymbol{u}_j^\eta \right\|_{L^2(\Omega)}^2 + \left\| \hat{\theta}^\eta - \mu_\theta^\eta \right\|^2 + \beta_{KL} \|\mu_{\boldsymbol{z}}^\eta\|^2 \right)$$

for the unsupervised scenario, and

$$L_{loss} = \frac{1}{S} \sum_{\eta=1}^{S} \left( \beta_d \sum_{j=1}^{n_{train}} \left\| \hat{\mathcal{G}}[\boldsymbol{f}_j^\eta; \hat{\theta}^\eta] - \boldsymbol{u}_j^\eta \right\|_{L^2(\Omega)}^2 + \left\| \hat{\theta}^\eta - \mu_\theta^\eta \right\|^2 + \beta_{KL} \|\mu_{\boldsymbol{z}}^\eta - \boldsymbol{b}^\eta\|^2 \right).$$

for the fully supervised scenario. Here, $\beta_d := \frac{\sigma_\theta^2}{\varpi^2 \Delta x^{2d_u}}$ and $\beta_{KL} := \sigma_\theta^2$ are treated as tunable hyperparameters. In the semi-supervised scenario, we incorporate partial supervision by equipping the above loss with a constraint following Locatello et al. (2020):

$$L_{loss} = \frac{1}{S} \sum_{\eta=1}^{S} \left( \beta_d \sum_{j=1}^{n_{train}} \left\| \hat{\mathcal{G}}[\boldsymbol{f}_j^\eta; \hat{\theta}^\eta] - \boldsymbol{u}_j^\eta \right\|_{L^2(\Omega)}^2 + \left\| \hat{\theta}^\eta - \mu_\theta^\eta \right\|^2 + \beta_{KL} \|\mu_{\boldsymbol{z}}^\eta\|^2 + \beta_{cls} L_c(\boldsymbol{z}^\eta, c(\beta^\eta)) \right)$$

with $\beta_{cls}$ a tunable parameter and $L_c(\boldsymbol{z}^\eta, c(\beta^\eta))$ the corresponding loss term based on provided information (e.g., $L_c$ is the cross-entropy loss when taking $c$ as the classification label of $\beta^\eta$).

Note that $\beta_{KL}$ is closely connected to the adjustable hyperparameter in $\beta-$VAE (Higgins et al., 2017). Taking a larger $\beta_{KL}$ encourages disentanglement (Locatello et al., 2019), while it also adds additional constraint on the implicit capacity of the latent bottleneck resulting in information loss (Burgess et al., 2018). On the other hand, the data reconstruction term forces the latent factors to contribute to the "reconstruction" of the more complicated solution field in a global perspective, and hence increasing $\beta_d$ is anticipated to alleviate information loss. In our empirical study, we demonstrate the interplay between these two tunable hyperparameters.

## 4 EXPERIMENTS

We assess DisentangO across various physics modeling and discovery datasets, denoting a model with latent dimension $n$ as DisentangO-$n$. Our evaluation focuses on several key aspects. Firstly, we demonstrate the capability of DisentangO in forward PDE learning, and compare its performance with a total of 14 relevant baselines. In particular, we select 8 NO baselines, i.e., FNO (Li et al., 2020c), UFNO (Wen et al., 2022), NIO (Molinaro et al., 2023), WNO (Tripura & Chakraborty, 2022), PIANO (Zhang et al., 2024), MetaNO (Zhang et al., 2023), FUSE (Lingsch et al., 2024) and its extension, as well as six non-NO baselines, i.e., CAMEL (Blanke &

Table 1: Test errors and the number of trainable parameters in experiment 1. Bold number highlights the best method.

| Models | #param | per-epoch time (s) | Test errors | |
| --- | --- | --- | --- | --- |
| | | | data | $\boldsymbol{z}$ (SC1) |
| DisentangO | 697k | 12.2 | 1.65% | **4.63%** |
| MetaNO | 296k | 9.8 | **1.59%** | - |
| NIO | 709k | 5.6 | - | 15.16% |
| FNO | 698k | 9.1 | 2.45% | 14.55% |
| UFNO | 720k | 21.2 | 7.61% | 11.23% |
| WNO | 672k | 183.5 | 8.09% | 9.95% |
| PIANO | 699k | 16.9 | 7.99% | 15.35% |
| CAMEL | 654k | 5.53 | 112.28% | - |
| SMDP | 671k | 5.12 | - | 17.76% |
| InVAErt | 707k | 0.1 | - | 5.16% |
| FUSE | 706k | 2.2 | - | 4.99% |
| FUSE-f | 707k | 11.4 | 16.33% | 6.19% |
| VAE | 698k | 2.8 | 49.97% | 16.34% |
| convVAE | 664k | 2.8 | 81.11% | 16.27% |
| $\beta$-VAE | 698k | 2.8 | 51.16% | 16.47% |

Lelarge, 2023), InVAErt (Tong et al., 2024), SMDP (Holzschuh et al., 2023), two VAE variants (Kingma & Welling, 2013) and $\beta$-VAE (Higgins et al., 2017). Secondly, we showcase the merits of DisentangO in inverse modeling for interpretable physics discovery. We perform parametric studies on the associated disentanglement parameters. Lastly, we provide interpretations of the disentangled parameters. Further details and an additional experiment for identifiability demonstration are provided in Appendices B and C.4, respectively.

### 4.1 SUPERVISED FORWARD AND INVERSE PDE LEARNING

We start by investigating DisentangO's capability in solving both forward and inverse PDE problems in a fully supervised setting. Specifically, we consider the constitutive modeling of anisotropic fiber-reinforced hyperelastic materials governed by the Holzapfel–Gasser–Ogden (HGO) model,

where the data-generating process is controlled by sampling the governing material parameter set $\{E, \nu, k_1, k_2, \alpha\}$ and the latent factors can be learned consequently in a supervised fashion. In this setting, the model takes as input the padded traction field and learns to predict the displacement.

As the number of latent factors is fixed for supervised learning in this experiment, we defer our ablation study to the second experiment. We report in Table 1 our experimental results. Note that only FNO, PIANO, FUSE-f, VAE and its variations are able to handle both forward and inverse problems, whereas MetaNO can only solve the forward problem, and NIO, InVAErt, and FUSE can only solve the inverse problem. With this caveat in mind, MetaNO achieves the best performance in forward PDE learning, with DisentangO performing comparably well and beating the third best model by 32.7% in accuracy. In terms of inverse modeling with full latent supervision (SC1), DisentangO is the only method that can hold the error well below 5%, outperforming the second best method by 7.2% and the second best joint (i.e., simultaneous forward and inverse) solver by 25.2%.

## 4.2 SEMI-SUPERVISED MECHANICAL MNIST BENCHMARK

We consider semi-supervised learning and apply DisentangO to the Mechanical MNIST (MMNIST) benchmark (Lejeune, 2020). MMNIST comprises 70,000 heterogeneous material specimens undergoing large deformation, each governed by a material model of the Neo-Hookean type with a varying modulus converted from the MNIST bitmap images. In our experiment, we take 500 images (with a 420/40/40 split for training/validation/test) and generate 200 loading/response data pairs per sample on a $29 \times 29$ structured grid, simulating uniaxial extension, shear, equibiaxial extension, and confined compression load scenarios. Since only partial knowledge is available for each image (i.e., the corresponding digit), we apply semi-supervised learning to the latent factors to classify the digits.

Table 2: Test errors and number of trainable parameters in experiment 2. DisentangO is abbreviated as DNO due to space limit. Bold number highlights the best method that can handle both forward and inverse settings.

| Models | DNO-2 | DNO-5 | DNO-10 | DNO-15 | VAE | $\beta$-VAE | MetaNO | PIANO |
|---|---|---|---|---|---|---|---|---|
| #param (M) | 0.66 | 0.97 | 1.49 | 2.02 | 2.02 | 2.02 | 0.35 | 2.05 |
| $\beta_d = 1$ | 12.82% | 9.56% | 7.36% | 6.29% | 16.34% | 17.13% | 2.68% | 13.73% |
| $\beta_d = 10$ | 11.51% | 9.16% | 6.62% | 5.95% | - | - | - | |
| $\beta_d = 100$ | 11.49% | 8.43% | 6.65% | **5.48%** | - | - | - | |
| $\beta_d = 1000$ | 11.62% | 8.22% | 6.50% | 5.80% | - | - | - | |

**Ablation study.** Firstly, we investigate DisentangO's predictability in forward PDE learning by comparing its performance to MetaNO (i.e., the base meta-learned NO model without the hypernetwork structure). As seen in Table 2, MetaNO achieves a forward prediction error of 2.68%, which serves as the optimal bound for DisentangO. As we increase the latent dimension in DisentangO from 2 to 15, the prediction error drops from 11.49% to 5.48% and converges to the optimal bound. Next, we study the role of the data loss term in disentanglement by gradually varying $\beta_d$ from $\beta_d = 1$ to $\beta_d = 1000$. In Table 2 we observe a consistent improvement in accuracy with the increase in $\beta_d$, where the boost in accuracy becomes marginal or slightly deteriorates beyond $\beta_d = 100$. We thus choose $\beta_d = 100$ as the best DisentangO model in this case. Besides offering an increased accuracy in forward modeling, the data loss term also enhances disentanglement as discussed in Section 3.2. This is evidenced by the illustration in Figure 3, where the unsupervised mutual information (MI) score that measures the amount of MI across latent factors consistently decreases in both DisentangO-2 and DisentangO-15 as we increase $\beta_d$ from $\beta_d = 1$ (solid lines) to $\beta_d = 100$ (dashed lines). On the contrary, the classification term poses a negative effect on disentanglement, as indicated by the increase in $\beta_{cls}$ leading to an increase in MI scores. This is reasonable because the classifier linearly combines all the latent factors to make a classification. The more accurate the classification, the stronger the correlation across latent factors. We then move on to study the effect of semi-supervised learning by comparing the model's performance with and without latent partial supervision. While the models without semi-supervision is slightly more accurate in that the forward prediction accuracy with $\beta_d = 1$ reaches 12.60% and 6.16% in DisentangO-2 and DisentangO-15, respectively, the latent scatterplot in Figure 2 reveals the inability of the DisentangO model without latent semi-supervision to acquire the partial knowledge of the embedded digits from data. In contrast, although the accuracy of DisentangO with latent semi-supervision slightly deteriorates due to the additional regularization effect from the classification loss term, it is able to correctly recognize the embedded digits and leverage this partial knowledge in disentangling meaningful latent factors.

**Comparison with additional baselines.** We compare DisentangO with additional baselines, i.e., VAE and $\beta$-VAE. We abandon the three NO-related baselines in this and the following experiments

as they are black-box approximators and do not possess any mechanism to extract meaningful information without supervision. In this context, even the least accurate DisentangO-2 model (i.e., with $\beta_d = 1$) with significantly fewer parameters outperforms the selected baselines in Table 2 by 21.5% and 25.2%, respectively, with the best performing DisentangO-15 model beating the baselines by 66.5% and 68.0% in accuracy, respectively. We do not proceed to compare the physics discovery capability between them as the baseline models are considerably inaccurate in forward prediction.

**Interpretable physics discovery.** To showcase the interpretability of DisentangO, we visualize its latent variables through latent traversal based on a randomly picked loading field as input in DisentangO-2, as displayed in Figure 6 of Appendix B.3. One can clearly see that the digit changes from "6" to "0" and then "2" from the top left moving down, and from "6" to "1" and then "7" moving to the right. Other digits are visible as well such as "7", "9", "4" and "8" in the right-most column. This corresponds well to the distribution of the latent clustering in Figure 2. More discussions on the interpretability of DisentangO-15 can be found in Figures 7 and 8 in Appendix B.3.

## 4.3 UNSUPERVISED HETEROGENEOUS MATERIAL LEARNING

We demonstrate DisentangO in unsupervised learning in the context of learning synthetic tissues that exhibit highly organized structures, with collagen fiber arrangements varying spatially. In this case, it is critical to understand the underlying low-dimensional disentangled properties in the hidden latent space of complex, high-dimensional microstructure, as inferred from experimental mechanical measurements. We generate two datasets, each containing 500 specimens and 100 loading/displacement pairs. The first dataset features variations in fiber orientation distributions using a Gaussian Random Field (GRF) and the second differs fiber angles in two segmented regions, separated by a centerline with a randomly rotated orientation.

Table 3: Test errors and number of trainable parameters in experiment 3. DisentangO is abbreviated as DNO due to space limit. Bold number highlights the best method for both forward and inverse settings.

| Models | #param (M) | Test error | |
|---|---|---|---|
| | | $\beta_d = 1$ | $\beta_d = 100$ |
| DNO-2 | 0.63 | 26.33% | 25.18% |
| DNO-5 | 0.94 | 17.51% | 15.75% |
| DNO-10 | 1.46 | 10.76% | 10.01% |
| DNO-15 | 1.98 | 7.11% | 7.02% |
| DNO-30 | 3.55 | 5.33% | **5.28%** |
| VAE | 3.55 | 61.10% | - |
| $\beta$-VAE | 3.55 | 57.04% | - |
| MetaNO | 0.32 | 2.67% | - |
| PIANO | 3.42 | 50.71% | |

We report in Table 3 our experimental results of DisentangO with different latent dimensions and data loss strength $\beta_d$, along with comparisons with the baseline model results. Consistent with the findings in the second experiment, increasing $\beta_d$ results in a boost in accuracy by comparing the rows in the table. The model's predictive performance also converges to the optimal bound of MetaNO as we increase the latent dimension from 2 to 30, where the model's prediction error improves from 25.18% to 5.28%. This significantly outperforms the best baseline model by 90.7%. On the other hand, the effect of the data loss term on disentanglement is further proved in Figure 4, where increasing $\beta_d$ leads to a decrease in MI between the latent factors, thus encouraging disentanglement. Lastly, we interpret the mechanism of the learned latent factors in DisentangO-3 via learning a mapping between the learned latent factors and the underlying material microstructure and subsequently performing latent traversal in each dimension. The results are shown in Figure 5, where the three latent factors manifest control on the border rotation between the two segments, the relative fiber orientation between the two segments, and the fiber orientation of the top segment, respectively.

## 5 CONCLUSION

We present DisentangO for disentangling latent physical factors embedded within black-box NO parameters. DisentangO leverages a hypernetwork-type NO architecture that extracts varying parameters of the governing PDE through a task-wise adaptive layer, and further disentangles these variations into distinct latent factors. By learning disentangled representations, DisentangO not only enhances physical interpretability but also enables robust generalization across diverse physical systems under different learning contexts. **Limitations:** Since the scalability of DisentangO is governed by the scalability of the NO backbone used for forward modeling, which has been extensively discussed in previous works, this study focuses on experiments involving high latent dimensions. Demonstrations on high-dimensional PDEs are beyond the scope of the current work.

ACKNOWLEDGMENTS

L. Zhang was partially funded by the NSF grant DMS 2436624 and the AFOSR grant FA9550-22-1-0197. Y. Yu would like to acknowledge the support by the National Institute of Health under award 1R01GM157589-01. Portions of this research were conducted on Lehigh University's Research Computing infrastructure, partially supported by NSF Award 2019035.

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

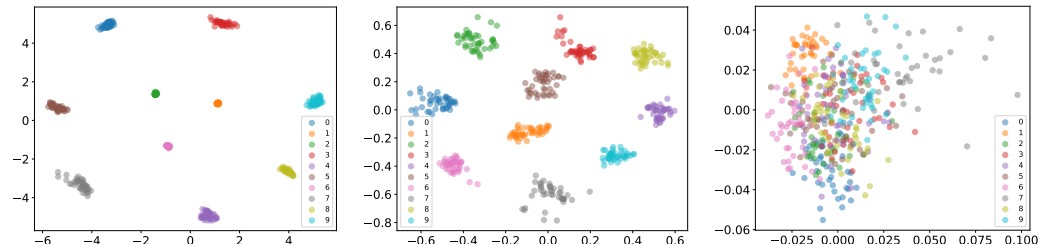

Figure 2: MMNIST scatterplot with DisentangO-2 and $\beta_d = 1$: left: $(\beta_{kl} = 1, \beta_{cls} = 100, \text{data error } 18.81\%)$, middle: $(\beta_{kl} = 10, \beta_{cls} = 10, \text{data error } 16.94\%)$, right: fully unsupervised DisentangO-2 without classification loss $(\beta_{kl} = 100, \text{data error } 12.65\%)$.

## A  IDENTIFIABILITY ANALYSIS

### A.1  PROOF OF THE MAIN THEOREMS

We first provide the proof of Theorem 1:

**Proof:** With equation 3.16, we have:
$$p_{\mathcal{G}[\boldsymbol{f};\theta]|\boldsymbol{f}} = p_{\hat{\mathcal{G}}[\boldsymbol{f};\hat{\theta}]|\boldsymbol{f}} \Leftrightarrow p_{\mathcal{G}_f(\theta)|\boldsymbol{f}} = p_{\hat{\mathcal{G}}_f(\hat{\theta})|\boldsymbol{f}} \Leftrightarrow p_{\theta|\boldsymbol{f}} = p_{\mathcal{G}_f^{-1}\circ\hat{\mathcal{G}}_f(\hat{\theta})|\boldsymbol{f}}.$$

Note that the parameter $\theta$ varies with the change of $\boldsymbol{b}$. Per the data generating process in equation 3.6, the distribution of $\boldsymbol{b}$ is invariant to $\boldsymbol{f}$. Therefore, the distribution of $\theta$ is also invariant to $\boldsymbol{f}$:
$$p_\theta = p_{\theta|\boldsymbol{f}} = p_{\mathcal{G}_f^{-1}\circ\hat{\mathcal{G}}_f(\hat{\theta})|\boldsymbol{f}} = p_{\mathcal{G}_f^{-1}\circ\hat{\mathcal{G}}_f(\hat{\theta})}, \ \forall \boldsymbol{f} \in \mathcal{F}.$$

Denoting $r := \mathcal{G}_f \circ \hat{\mathcal{G}}_f^{-1}$, it is the transformation between the true $\theta$ and the estimated one, and it is invertible and invariant with respect to $\boldsymbol{f}$.

We proceed to derive the relation between $\boldsymbol{z}$ and $\hat{\boldsymbol{z}}$: since $\theta = r(\hat{\theta})$, with the invertibility assumption $\theta = \mathcal{H}^{-1}(\boldsymbol{z})$ and $\hat{\theta} = \hat{\mathcal{H}}^{-1}(\hat{\boldsymbol{z}})$, we obtain:
$$\boldsymbol{z} = \mathcal{H}(\theta) = \mathcal{H} \circ r(\hat{\theta}) = \mathcal{H} \circ r \circ \hat{\mathcal{H}}^{-1}(\hat{\boldsymbol{z}}).$$

Denoting $h := \mathcal{H}\circ r\circ\hat{\mathcal{H}}^{-1}$, it is the transformation between the true latent variable and the estimated one, and it is invertible because $r$, $\mathcal{H}$ and $\hat{\mathcal{H}}$ are all invertible. □

We now show the proof of Theorem 2.

**Proof:** With the independence relation assumption, we have
$$p_{\boldsymbol{z}|\boldsymbol{b}}(\boldsymbol{z}|\boldsymbol{b}) = \prod_i p_{z_i|\boldsymbol{b}}(z_i), \ p_{\hat{\boldsymbol{z}}|\boldsymbol{b}}(\hat{\boldsymbol{z}}|\boldsymbol{b}) = \prod_i p_{\hat{z}_i|\boldsymbol{b}}(\hat{z}_i) .$$

Denoting $\hat{q}_i := \log p_{\hat{z}_i|\boldsymbol{b}}$, it yields:
$$\log p_{\boldsymbol{z}|\boldsymbol{b}}(\boldsymbol{z}|\boldsymbol{b}) = \sum_i q_i(z_i, \boldsymbol{b}), \ \log p_{\hat{\boldsymbol{z}}|\boldsymbol{b}}(\hat{\boldsymbol{z}}|\boldsymbol{b}) = \sum_i \hat{q}_i(\hat{z}_i, \boldsymbol{b}) .$$

With the change of variables we have
$$p_{\boldsymbol{z}|\boldsymbol{b}} = p_{h(\hat{\boldsymbol{z}})|\boldsymbol{b}} = p_{\hat{\boldsymbol{z}}|\boldsymbol{b}} \cdot |J_{h^{-1}}| \Leftrightarrow \sum_i q_i(z_i, \boldsymbol{b}) + \log|J_h| = \sum_i \hat{q}_i(\hat{z}_i, \boldsymbol{b}) ,$$

where $|J_{h^{-1}}|$ stands for the absolute value of the Jacobian matrix determinant of $h^{-1}$. Differentiating the above equation twice with respect to $\hat{z}_k$ and $\hat{z}_q$, $k \neq q$, yields
$$\sum_i \left( \frac{\partial^2 q_i(z_i, \boldsymbol{b})}{\partial z_i^2} \frac{\partial z_i}{\partial \hat{z}_k} \frac{\partial z_i}{\partial \hat{z}_q} + \frac{\partial q_i(z_i, \boldsymbol{b})}{\partial z_i} \frac{\partial^2 z_i}{\partial \hat{z}_k \partial \hat{z}_q} \right) + \frac{\partial^2 \log|J_h|}{\partial \hat{z}_k \partial \hat{z}_q} = 0 . \tag{A.1}$$

To show the identifiability, one can rewrite the Jacobian $J_h$ as:
$$J_h = \left[ \frac{\partial \boldsymbol{z}}{\partial \hat{\boldsymbol{z}}} \right] .$$

The invertibility results shown in Theorem 1 indicates that it is full rank. Next, we will use the linear independence assumption to show that there exists one and only one non-zero component in each row of $\frac{\partial \boldsymbol{z}}{\partial \hat{\boldsymbol{z}}}$.

Taking $\boldsymbol{b} = \boldsymbol{b}^0, \cdots, \boldsymbol{b}^{2d_z}$ in equation A.1 and subtracting them from each other, we have

$$\sum_{i=1}^{d_z} \left( \left( \frac{\partial^2 q_i(z_i, \boldsymbol{b}^j)}{\partial z_i^2} - \frac{\partial^2 q_i(z_i, \boldsymbol{b}^0)}{\partial z_i^2} \right) \frac{\partial z_i}{\partial \hat{z}_k} \frac{\partial z_i}{\partial \hat{z}_q} + \left( \frac{\partial q_i(z_i, \boldsymbol{b}^j)}{\partial z_i} - \frac{\partial q_i(z_i, \boldsymbol{b}^0)}{\partial z_i} \right) \frac{\partial^2 z_i}{\partial \hat{z}_k \partial \hat{z}_q} \right) = 0 \,,$$

where $j = 1, \cdots, 2d_z$. With the linear independence condition for $w$, this is a $2d_z \times 2d_z$ linear system, and therefore the only solution is

$$\frac{\partial z_i}{\partial \hat{z}_k} \frac{\partial z_i}{\partial \hat{z}_q} = 0 \,, \qquad \frac{\partial^2 z_i}{\partial \hat{z}_k \partial \hat{z}_q} = 0 \,,$$

for $i = 1, \cdots, d_z$. The first part implies that, for the $i-$th row of the Jacobian matrix $J_h$, we have $\frac{\partial z_i}{\partial \hat{z}_k} \neq 0$ for at most one element $k \in \{1, \cdots, d_z\}$, hence $\boldsymbol{z}$ is identifiable up to permutation and component-wise invertible transformation. $\qquad \square$

## A.2 FURTHER DISCUSSION ON THE ASSUMPTIONS

Herein, we provide additional discussion on the validity and empricial validation for Assumptions 1-4.

As seen in the proof above, Assumptions 1 (Density Smoothness and Positivity) and 2 (Invertibility) are required to guarantee that there exists a smooth and injective mapping $h := \mathcal{H} \circ r \circ \hat{\mathcal{H}}^{-1}$, from the ground-truth latent embedding $\boldsymbol{z}$ to the learned embedding $\hat{\boldsymbol{z}}$. Furthermore, the smoothness assumption further makes it feasible to take derivatives of $\boldsymbol{z}$ with respect to $\hat{\boldsymbol{z}}$, which supports the permutation-wise identifiability proof for Theorem 2. Here, we note that the smoothness assumption may possibly be relaxed to $C^2$. Assumptions 3 (Conditional Independence) and 4 (Linear Independence) are needed to show that the Jacobian of $h$ has one and only one non-zero component for each column. Without these assumptions, it is possible that the data from different $\boldsymbol{b}$ lack variability.

While the first three assumptions are common in many VAE architectures, the last assumption is plausible for many real-world data distributions. For instance, when the prior on the latent variables $p(\boldsymbol{z}|\boldsymbol{b})$ is conditionally factorial, where each element $z_i$ has a univariate exponential family distribution given conditioning variable $\boldsymbol{b}$:

$$p(\boldsymbol{z}|\boldsymbol{b}) = \prod_i \frac{Q_i(z_i)}{Z_i(\boldsymbol{b})} \exp \left[ \sum_{j=1}^k T_{i,j}(z_i) \lambda_{i,j}(\boldsymbol{b}) \right] \,,$$

where $Q_i$ is the base measure, $Z_i(\boldsymbol{b})$ is the normalizing constant, $T_{i,j}$ are the sufficient statistics, and $\lambda_{i,j}$ are the corresponding parameters depending on $\boldsymbol{b}$. This exponential family has universal approximation capabilities. Additionally, we note that this distribution is conditionally independent with

$$q_i = \log(Q_i(z_i)) - \log(Z_i(\boldsymbol{b})) + \left[ \sum_{j=1}^k T_{i,j}(z_i) \lambda_{i,j}(\boldsymbol{b}) \right] \,,$$

and the linear independence indicates that the matrix formed by

$$\omega(\boldsymbol{z}, \boldsymbol{b}^\eta) - \omega(\boldsymbol{z}, \boldsymbol{b}^0) = \left( \mathbf{T}'(\lambda(\boldsymbol{b}^\eta) - \lambda(\boldsymbol{b}^0)), \mathbf{T}''(\lambda(\boldsymbol{b}^\eta) - \lambda(\boldsymbol{b}^0)) \right) \,, \eta = 1, \cdots, S$$

has full rank $2d_z$.

In the fully supervised case, the conditional independence and linear independence assumptions are automatically guaranteed by picking proper distributions $p(\boldsymbol{z}|\boldsymbol{b})$ when designing the VAE architecture. In the semi-supervised and unsupervised cases, one can also validate these assumptions when the true values of $\boldsymbol{b}$ is given on some tasks, by inferring an empirical distribution of $p(\boldsymbol{z}|\boldsymbol{b})$ from these tasks. To investigate such a capability, in the additional synthetic experiment in Appendix C.4, we consider an unsupervised setting, estimate $p(\boldsymbol{z}|\boldsymbol{b})$ from the trained model, and check the linear independence condition by calculating the vector in equation 3.15 for each task $\boldsymbol{b}^\eta$ and forming an $S \times 2d_z$ matrix from all tasks. When the rank of this matrix is $2d_z$, it means that we can select $2d_z + 1$ tasks from them with sufficient variability, such that the linear independence condition is satisfied. As a demonstration, in Appendix C.4 we validate this assumption on a synthetic dataset and show that the identifiability can be largely achieved with Gaussian distributions of distinct means and variances.

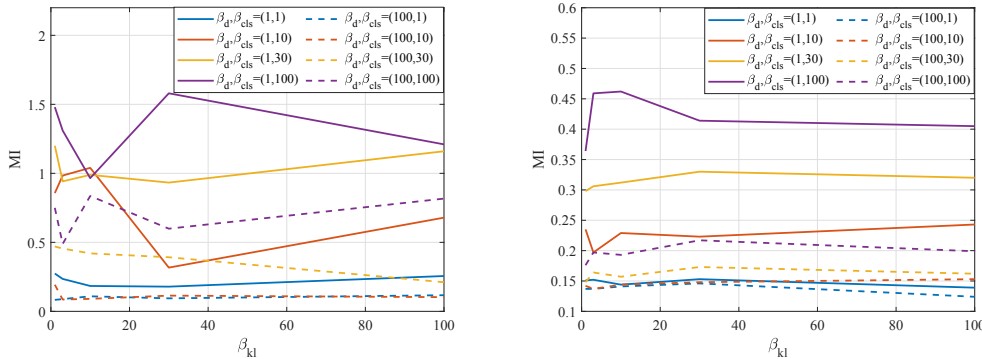

Figure 3: MMNIST unsupervised scores against $\beta_d$ with DisentangO-2 (left) and DisentangO-15 (right). By comparing $\beta_d = 1$ (solid lines) with $\beta_d = 100$ (dashed lines), increasing $\beta_d$ forces the latent factors to maximize the contained information and in turn decreases MI, thus encouraging disentanglement.

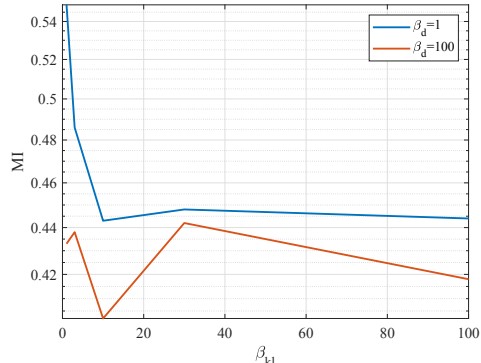

Figure 4: Unsupervised MI score against $\beta_{kl}$ with DisentangO-2 in heterogeneous material learning.

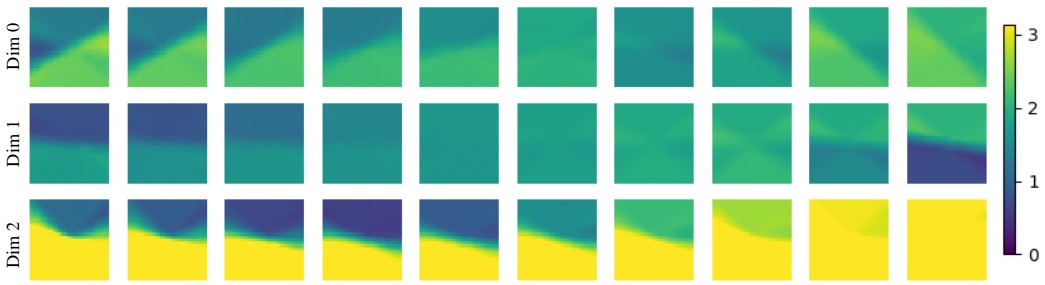

Figure 5: Latent traversal of DisentangO-3 in unsupervised heterogeneous material learning, where the three latent dimensions control the border rotation between the two segments (top), the relative fiber orientation between the two segments (middle), and the fiber orientation of the top segment (bottom), respectively. Legend indicates fiber orientation ranging from 0 to $\pi$.

# B ADDITIONAL EXPERIMENTAL DETAILS, RESULTS AND DISCUSSION

We provide additional details in training and baseline models, as well as more results as a supplement of Section 4.

## B.1 TRAINING DETAILS

In all experiments, we adopt the Adam optimizer for optimization and use a $n$-layer DisentangO model, where $n = 8$ in the first experiment and $n = 16$ in the second and third experiments due to increased complexity. For fair comparison across different models, we tune the hyperparameters,

including the learning rates, the decay rates, and the regularization parameters, to minimize the validation loss. Experiments are conducted on a single NVIDIA Tesla A100 GPU with 40 GB memory. A pseudo algorithm of all three scenarios is summarized in Algorithm 1.

---

**Algorithm 1** A pseudo algorithm of DisentangO.

---

1: Denote data reconstruction loss $L_{data}$, task-wise NO parameter reconstruction loss $L_{recon}$, KL loss $L_{KL}$ and semi-supervision loss $L_{semi}$ as:

$$L_{data} = \frac{1}{S}\sum_{\eta=1}^{S}\left[-E(\log(p(\boldsymbol{u}|\boldsymbol{f},\theta^\eta)))\right] , \quad L_{recon} = \frac{1}{S}\sum_{\eta=1}^{S}\left[-E_{q(\boldsymbol{z}^\eta|\theta^\eta)}\log p(\theta^\eta|\boldsymbol{z}^\eta)\right] ,$$

$$L_{KL} = \frac{1}{S}\sum_{\eta=1}^{S}\left[D_{KL}\left(q(\boldsymbol{z}^\eta|\theta^\eta)||p(\boldsymbol{z}^\eta)\right)\right] , \quad L_{semi} = L_c(\boldsymbol{z}^\eta, c(\boldsymbol{b}^\eta)) .$$

2: **SC1: Supervised/SC3: Unsupervised**
3: The total loss is comprised of the data loss, the NO parameter reconstruction loss, and the KL loss: $L_{loss} = \beta_d L_{data} + L_{recon} + \beta_{KL} L_{KL}$.
4: **SC2: Semi-supervised**
5: The total loss is comprised of the data loss, the NO parameter reconstruction loss, the KL loss, and a semi-supervised loss: $L_{loss} = \beta_d L_{data} + L_{recon} + \beta_{KL} L_{KL} + \beta_{cls} L_{semi}$.

---

## B.2 Supervised forward and inverse PDE learning

The parameter of each baseline is given in the following, where the parameter choice of each model is selected by tuning the number of layers and the width (channel dimension), keeping the total number of parameters on the same magnitude.

- MetaNO: We use a 8-layer IFNO model with the lifting layer as the adaptive layer. We keep the total number of parameters in MetaNO the same as the number of parameters used in forward PDE learning in DisentangO.

- NIO: We closely follow the setup in Molinaro et al. (2023), where two neural operators (DeepONet and FNO) are stacked together to realize the operator-to-function intuition. The first operator maps multiple solution functions to a set of representations (which can be seen as an analog of eigenfunctions), and the second operator infers the underlying parameter field from the mixed representations. As NIO requires the solution field as input, it cannot be used as a forward solver. Hence, NIO only solves the inverse PDE problem, and it can only be applied to the fully supervised setting. Specifically, we use four convolution blocks as the encoder for the branch net and a fully connected neural network with two hidden layers of 256 neurons as the trunk net, with the number of basis functions set to 50. For the FNO part, we use one Fourier layer with width 32 and modes 8, as suggested in Molinaro et al. (2023).

- FNO: Since FNO is originally designed as a function-to-function mapping, we consider the inverse optimization procedure following Lee et al. (2024), and develop a two-phase process to solve the forward and inverse problems sequentially. In the first phase, we construct the forward mapping from the loading field $\boldsymbol{f}$ and the ground-truth material parameter $\boldsymbol{b}$ to the corresponding solution $\boldsymbol{u}$ as: $\mathcal{G}^{FNO}[\boldsymbol{f},\boldsymbol{b};\theta^{FNO}](\boldsymbol{x}) = \boldsymbol{u}(\boldsymbol{x})$. This can be seen as an analog of the forward solution operator $\mathcal{G}$ in our setting. Then, with the trained FNO as a surrogate for the forward solution operator, we fix its NN parameters $\theta^{FNO}$, and use it together with gradient-based optimization to solve for the optimal material parameters as an inverse solver. Specifically, given a set of loading/solution data pairs $\{(f_i, u_i)\}_{i=1}^{N}$, we start from a random guess of the underlying material parameters (typically chosen as the average of all available instances of material parameters for fast convergence), and minimize the difference between the predicted displacement field from FNO and the ground-truth one:

$$b^* = \operatorname{argmin}_b \sum_{i=1}^{N}\|u_i - \mathcal{G}^{FNO}[f_i, b; \theta^{FNO}]\|^2.$$

As the FNO parameters are fixed, we can back propagate this loss and optimize the input material parameters in an iterative fashion. We adopt a 4-layer FNO with width 26 and

modes 8. For the forward model, in addition to the loading field and the coordinates as input, we also concatenate the ground-truth material properties to form the final input. For the inverse model, we employ an iterative gradient-based optimization to solve for the optimal material parameters for physics discovery. For fair comparison, in terms of the averaged per-epoch runtime, we report the sum of both the forward and inverse solvers. The averaged per-epoch runtimes for the forward solver and the inverse solver are 6.2 seconds and 2.9 seconds, respectively, accounting for the total per-epoch runtime of 9.1 seconds in Table 1.

- UFNO: The 2D U-FNO model extends the Fourier Neural Operator (FNO) architecture by incorporating U-Net-style skip connections. The network consists of six spectral convolution blocks, each combining a global Fourier operator and a local $1 \times 1$ convolution. In the later three layers, additional skip-enhanced feature extraction is provided by U-Net blocks. The input consists of spatial features $f(x, y)$, concatenated with coordinate embeddings $(x, y)$ and 5 material parameters. The input is lifted to a higher-dimensional latent space via a linear layer of size $[8, 20]$. Each of the six layers performs the following composite operation:

$$v_{j+1} = \text{ReLU}(\mathcal{F}_j(v_j) + \mathcal{W}_j(v_j) + \mathcal{U}_j(v_j)),$$

where $\mathcal{F}_j$ is a spectral convolution layer (2D Fourier transform, mode truncation, linear transformation, and inverse transform), $\mathcal{W}_j$ is a $1 \times 1$ convolution, and $\mathcal{U}_j$ is a U-Net block (included only in layers 4 to 6). After all six layers, the output is projected through two fully connected layers of sizes $[20, 128]$ and $[128, 2]$ to produce the final 2-channel output.

- WNO: The Wavelet Neural Operator (WNO) consists of 4 stacked wavelet kernel integral layers. The input is first lifted to a high-dimensional representation using a fully connected layer of size $[8, 27]$. Each of the 4 layers combines a learned local operator $W$ (implemented as $1 \times 1$ convolution) with a non-local wavelet-based integral operator $K$ (via continuous wavelet transform). The architecture follows: The input $f(x, y)$ is first augmented by concatenating it with the positional grid coordinates $(x, y)$ and additional 5 material parameters. This augmented input is then lifted into a higher-dimensional space via a fully connected layer that maps the input to a hidden width of `width`. The core of the architecture consists of 4 wavelet-based layers, the output is projected back to the desired output space using two linear layers of sizes $[27, 128]$ and $[128, 41 * 41 * 2]$. Mish activations are applied after each intermediate layer except the last projection layer.

- CAMEL: The architecture consists of two networks, V-Net and C-Net

$$G(f(x); \theta, w) = c((f(x); \theta) + w^\top v(f(x); \theta). \tag{B.1}$$

The score network $V_\phi$ is a 5-layer MLP with width 320 and Tanh activations, with layer sizes $[d, 320, 320, 320, 320, r]$ The coefficient network $c_\theta$ is a 3-layer MLP with width 128 and Tanh activations, with layer sizes $[d, 128, 128, 1]$,, where $d$ is the size of the flatten $f(x)$, and $r = 5$

- SMDP: Since we do not have an explicit physical operator to evaluate $P^{-1}(z, f(x))$, we use only the score network, implemented as a 6-layer MLP with layer sizes $[128, 512, 256, 128, 64, 32, d_z]$, where $d_z$ denotes the dimension of the latent space.

- PIANO: We use the ground-truth material properties as the physical-invariant embedding. The loading field is first processed by a lifting layer, while the material parameters are passed through an attention module implemented as a 3-layer MLP with hidden dimension 32. The outputs of the lifting layer and the attention module are then concatenated and fed into a convolutional layer with width 26, followed by a 3-layer IFNO with width 26 and modes 8. For both the forward and inverse solution procedures, we adopt the same settings as in the FNO baseline.

- InVAErt: We directly take the InVAErt implementation from Lingsch et al. (2024) and define the encoder, the VAE encoder and the decoder as 4-layer MLPs with hidden dimension 96 and silu activation function.

- FUSE: For the forward model, we take three Fourier layers in addition to the first band-limited lifting layer that increases the dimension of the parameters and performs an inverse Fourier transform. On the other hand, the inverse model maps the functional input to the parameter space by employing a concatenation of two Fourier layers and a band-limited forward Fourier transform that generates a fixed-size latent representation of the

input function, followed by a flow-matching posterior estimation (FMPE) that maps to the final parameter output. All Fourier layers have a latent width of 32 and 8 modes, while the FMPE flow has 4 layers of width 360.

- FUSE-f: Since the original FUSE model assumes a constant loading field, it cannot handle situations where the input loading field changes that create multiple instances of a PDE system. We therefore create a FUSE variation (denoted as "FUSE-f") that takes a concatenation of both the displacement field and the loading field as input to the inverse model. For the forward model, we concatenate the loading field to the output of the first band-limited lifting layer and subsequently use a one-layer MLP to map it back to the original dimension. All other settings are the same as the original FUSE baseline.

- VAE: We use a 2-layer MLP of size [1681, 136, 30] as the encoder, and another 2-layer MLP of size [30, 136, 1681] as the decoder, with the size of the bottleneck layer being 30.

- convVAE: We use a convolutional layer with 136 kernels of size $3 \times 3$ with a stride of 2 pixels and a fully connected layer of size [59976, 5] as the encoder, and a fully connected layer of size of [5, 59976] and a transposed convolutional layer with 2 kernels of size $3 \times 3$ as the decoder.

- $\beta$-VAE: The parameter choice of the $\beta$-VAE baseline is the same as the VAE baseline, except that we tune the $\beta$ hyperparameter.

**Model-agnostic architecture.** The proposed DisentangO architecture is model-agnostic and can incorporate any neural operator as the forward solver, including FNO, UFNO, and WNO. This flexibility stems from DisentangO's design, which only requires designating the lifting layer as the task-wise adaptation layer and adding the VAE structure for latent disentanglement (i.e., the encoder and the first decoder in Figure 1). DisentangO's overall performance scales directly with the underlying neural operator's capabilities. As long as the chosen NO achieves comparable forward prediction accuracy to IFNO, DisentangO will maintain similar performance, since (1) the upper bound of forward prediction accuracy depends on the base neural operator, and (2) the inverse prediction accuracy depends on how well the latent factors are encoded in the task-wise lifting layer parameters. We select IFNO for three key advantages: (1) parameter efficiency: the layer-independent parameter setting significantly reduces trainable parameters compared to alternatives, (2) theoretical guarantees: IFNO is a provably universal approximator as PDE solution operators, and (3) natural integration: seamless compatibility with MetaNO's lifting layer approach for task-wise adaptation. If the IFNO part is replaced with other neural operators such as the considered baselines of FNO, UFNO, and WNO as in Table 1, the forward prediction accuracy will decrease (cf. the data test errors in Table 1), meaning that the upper bound of DisentangO's forward prediction accuracy will decrease. Additionally, the inverse prediction accuracy (i.e., the z test error in Table 1) will likely drop as well, because the degraded forward prediction accuracy typically indicates degraded encoding of latent information in the lifting layer, which will negatively impact the inverse modeling even if the VAE part can reconstruct the lifting layer parameters perfectly. As IFNO performs the best in forward prediction and requires the least amount of trainable parameters, we choose IFNO as the forward prediction backbone in DisentangO.

To demonstrate DisentangO's compatibility with alternative neural operators, we replace IFNO with UFNO in the DisentangO architecture. The performance comparison is presented in Table 4.

Table 4: Test errors and the number of trainable parameters in experiment 1. Bold number highlights the best method.

| Models | #param | per-epoch time (s) | Test errors | |
|---|---|---|---|---|
| | | | data | $z$ (SC1) |
| DisentangO | 697k | 12.2 | 1.65% | **4.63%** |
| MetaNO | 296k | 9.8 | **1.59%** | - |
| MetaUFNO | 304k | 11.4 | 3.55% | - |
| DisentangO-UFNO | 688k | 14.2 | 6.61% | 9.85% |
| UFNO | 720k | 21.2 | 7.61% | 11.23% |

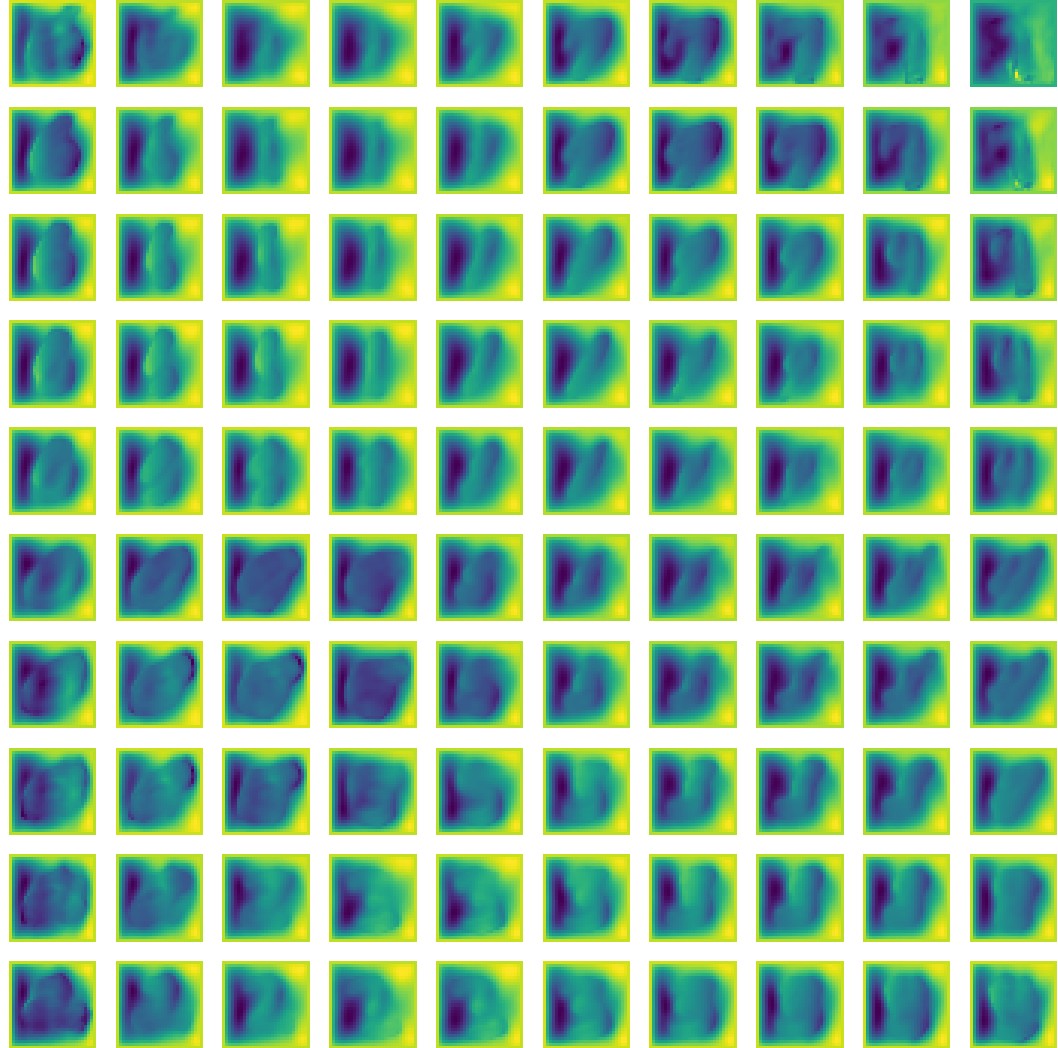

Figure 6: MMNIST latent traversal based on a randomly picked loading field as DisentangO-2 input.

## B.3 MECHANICAL MNIST BENCHMARK

We demonstrate the latent traversal based on a randomly picked loading field as DisentangO-2 input. One can clearly see that the digit changes from "6" to "0" and then "2" from the top left moving down, and from "6" to "1" and then "7" moving to the right. Other digits are visible as well such as "7", "9", "4" and "8" in the right-most column. This corresponds well to the distribution of the latent clustering in Figure 2. We also provide two exemplary MMNIST latent interpretations of the learned DisentangO-15 in Figures 7 and 8.

## B.4 LATENT VISUALIZATION AND PHYSICAL INTERPRETATION

As the hidden parameter field $b^\eta$ of the PDE is generally not accessible, especially in the semi-supervised or unsupervised settings, one cannot directly reconstruct $b^\eta$ from the learned latent variables $z$. Through disentangled representation learning, the latent variables $z$ are anticipated to contain critical information of $b^\eta$, but there does not exist a direct and explicit mapping from $z$ to $b^\eta$. As a result, one cannot directly reconstruct $b^\eta$ from $z$. However, there are several tricks one can play with to obtain meaningful interpretations from the learned $z$. On one hand, one can feed a randomly selected input into DisentangO and manually define a desired $z$ in the latent space and perform a forward pass using the trained DisentangO. $b^\eta$ can then be visualized from the output, as demonstrated

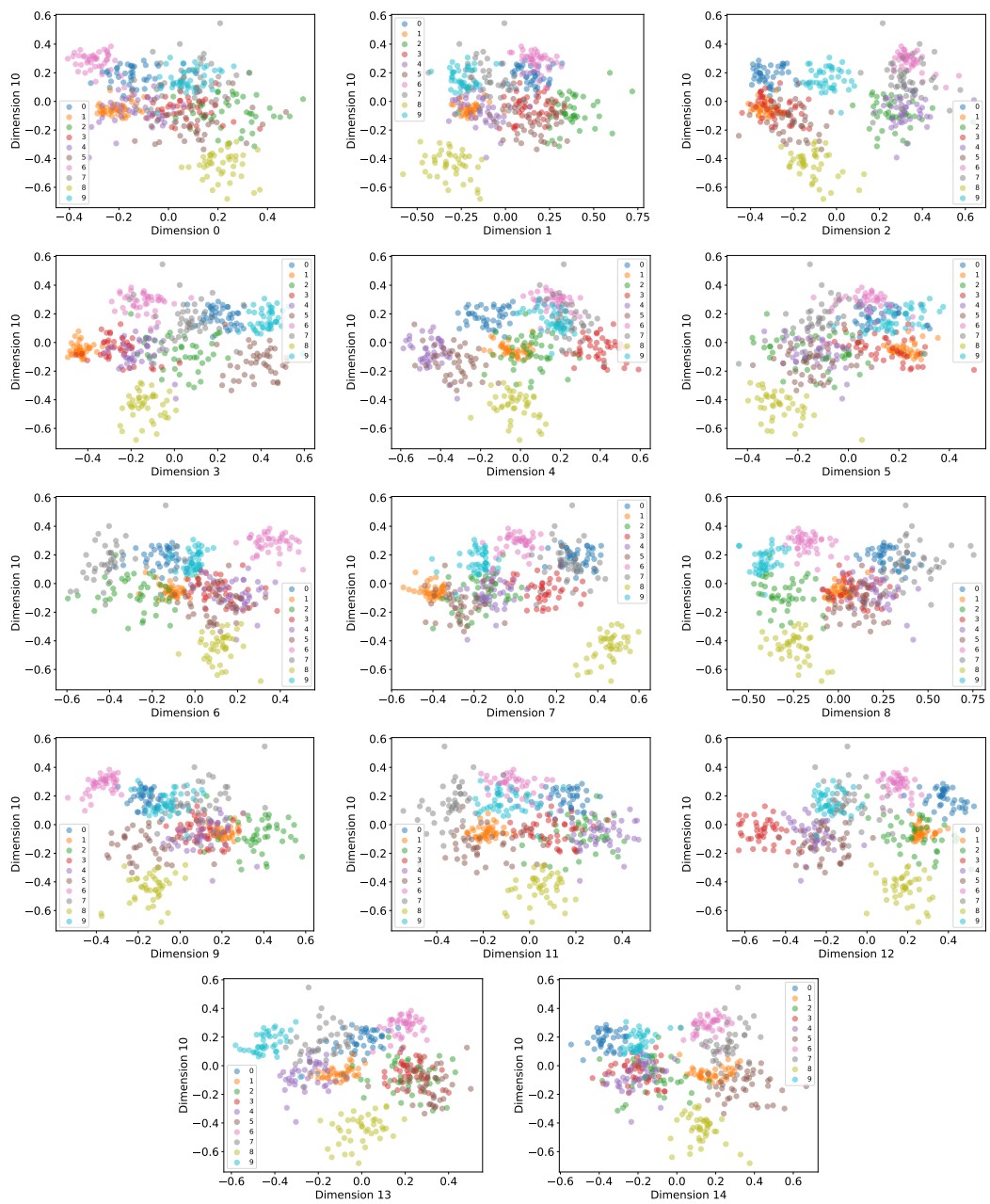

Figure 7: Exemplary MMNIST latent interpretation of DisentangO-15: dimension 10 encodes the information for digit '8', as is evidenced by the fact that all $y < -0.4$ regions on the scatterplots contain only digit '8'.

in Figure 6 in the MMNIST experiment. On the other hand, one can also train a simple MLP and construct a mapping from $z$ to $b^\eta$, provided that $b^\eta$ is available. With this mapping at hand, one can then traverse $z$ and visualize what each dimension of $z$ controls. This is demonstrated in Figure 5 in the third experiment.

## B.5 FURTHER DISCUSSION

**Probabilistic latent space.** We use a variational (probabilistic) encoder rather than a deterministic map for two reasons:

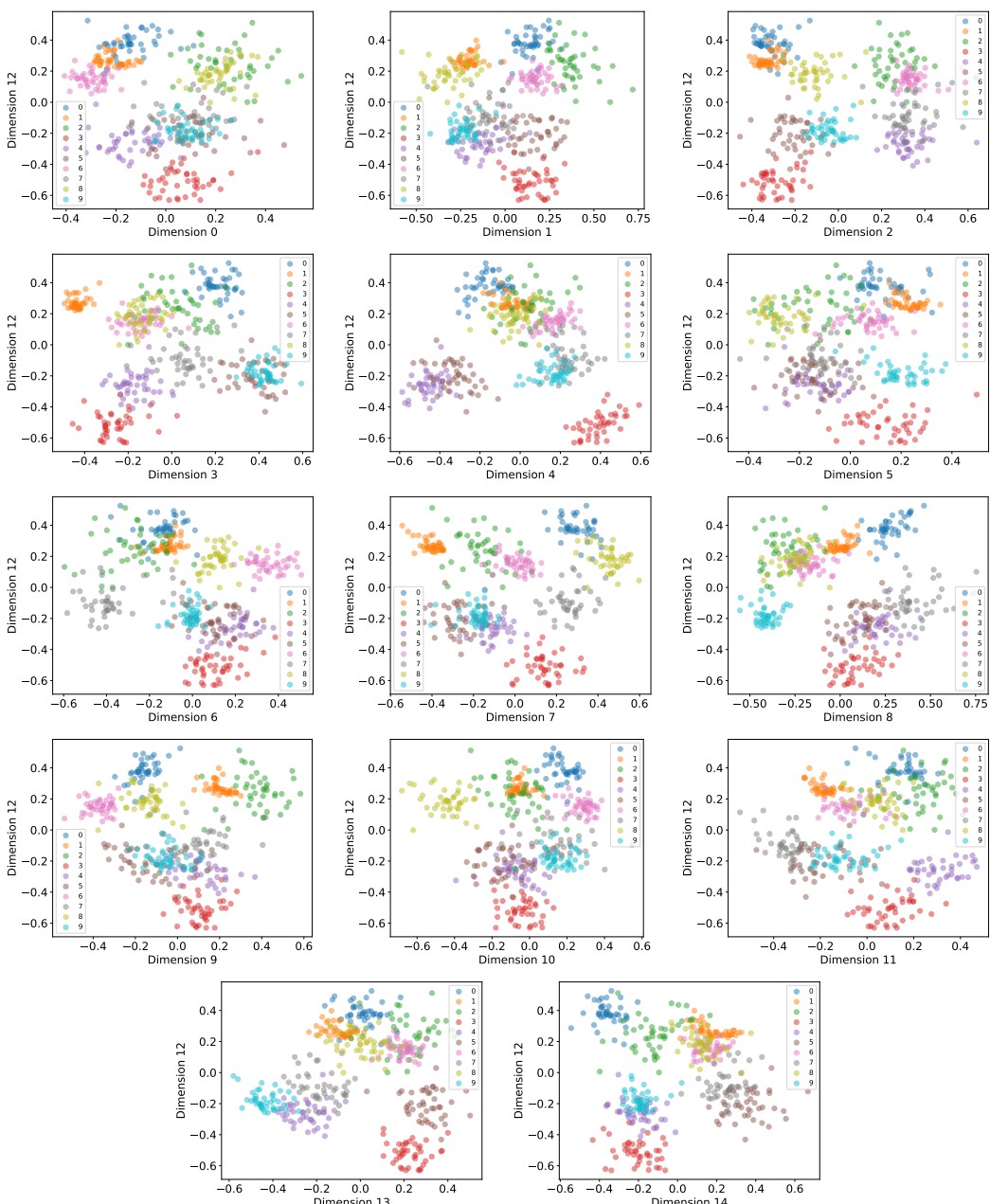

Figure 8: Exemplary MMNIST latent interpretation of DisentangO-15: dimension 12 encodes the information for digit '3', as is evidenced by the fact that all $y < -0.45$ regions on the scatterplots contain only digit '3'.

- Regularization and stability of the inverse map. The inverse operator in ill-posed PDE settings is highly sensitive: small perturbations in the solution $u$ can correspond to large changes in the inferred parameters. The VAE's KL term imposes a distributional prior on the latent variables, preventing the encoder from collapsing to arbitrarily sharp or unstable mappings. This yields a smoother, more stable inverse that generalizes better under noise and limited data.

- Identifiability and disentanglement. The identifiability results we reference (e.g., up to component-wise invertible transformations) rely on the latent distribution having a simple, factorized prior. The VAE provides exactly this structure: a normalized, independent latent prior that constrains the representation and supports the theoretical guarantees.

**Scalability.** Scaling interpretable neural operators to complex, high-dimensional PDEs is critical for broader impact. Our work is structured to first establish the foundational capabilities of DisentangO in controlled settings where disentanglement and representation quality can be rigorously verified. Notably, our work tackles scenarios with partially or fully unknown physics: the second experiment demonstrates DisentangO's ability to disentangle and identify latent codes when only partial knowledge is available, and the third experiment aims to mimic a real-world fully unsupervised scenario where the physical parameters are entirely unknown, yet DisentangO successfully extracts interpretable latent factors. DisentangO is designed to be inherently scalable for several key reasons: (1) Modular architecture: DisentangO inherits the computational efficiency of the underlying neural operator backbone (IFNO in our implementation), which supports batched, GPU-accelerated inference in high dimensions. (2) Constant latent dimension: the latent code dimensionality remains low regardless of spatial resolution or system complexity, enabling practical inverse inference even for complex systems. (3) Linear scaling: the latent inference network scales linearly with the number of PDE instances, independent of spatial dimensionality. (4) Plug-and-play design: DisentangO can incorporate any state-of-the-art neural operator as its backbone, inheriting its scalability properties. In essence, the scalability of DisentangO is governed by the scalability of the neural operator used for forward modeling. As long as the NO component performs well and the training data spans sufficient variability in physical system parameters, DisentangO remains applicable.

**Encoding $b$ in lifting layers.** Our choice to encode the parameters $b$ in the lifting layers $\theta_P$ is justified through three complementary perspectives. (1) Theoretical expressivity: as demonstrated in MetaNO (Zhang et al., 2023), varying only the lifting layer across tasks is sufficient to universally approximate a wide class of parametric operators. DisentangO inherits this universal approximation property while maintaining a meta-learning framework where the core iterative Fourier layers serve as a shared meta-operator across tasks. (2) Representation identifiability: restricting adaptation to the lifting layer significantly enhances the identifiability of latent representations, a critical requirement for disentangled inverse learning. When adaptation is allowed in deeper layers, the mapping between latent codes and operators becomes more diffuse and less interpretable. (3) Empirical validation: our experiments across diverse PDE types confirm that this design choice enables DisentangO to: (a) accurately model task-specific operators, (b) recover interpretable physical factors, and (c) generalize to unseen parameter combinations and interpolation tasks.

**Interpretability beyond parameter reconstruction.** Our approach provides interpretability at the level of identifiable physical factors, which is often more meaningful than raw parameter reconstruction. Many physical parameters are inherently non-identifiable from observational data alone (e.g., material properties in regions with zero stress). Our method identifies the factors that actually govern system behaviors. Moreover, our experiments demonstrate clear interpretable patterns: (1) MMNIST (Figure 6): latent traversal reveals smooth transitions between digit patterns, showing the method captures meaningful morphological variations, (2) Heterogeneous materials (Figure 5): each latent dimension controls specific physical aspects: border rotation, relative fiber orientation, and segment-wise fiber orientation, (3) Semi-supervised results (Figure 2): latent factors successfully cluster different digit classes without explicit supervision. Our theoretical results on component-wise identifiability (Theorems 1 and 2) guarantee that the learned factors $\hat{z}$ correspond to true generative factors $z$ up to invertible transformations. This provides theoretical grounding for interpretability claims. In terms of practical utility, even without exact $b$ reconstruction, the disentangled factors can enable system classification and similarity assessment, design space exploration through latent traversal, transfer learning to new parameter regimes, and anomaly detection in physical systems.

**Quantitative interpretability metric.** The evaluation of interpretability depends critically on the availability of ground truth, which varies across our three practical scenarios, i.e., supervised, semi-supervised, and unsupervised. In the fully supervised setting corresponding to experiment 1, When true generative factors are available, we use quantitative metrics. The interpretability metric is the latent supervision test error (cf. Table 1), which directly measures how well the learned latent variables $z$ recover the true physical parameters $b$. In the semi-supervised setting corresponding to experiment 2, the goal of interpreting physics is to discover the underlying microstructure governing the deformation. Under this setting, since the true generative factors are not available, one cannot use any closed-form metric to evaluate interpretability. However, since the ground-truth microstructure generation is controlled by MNIST digits, successful disentanglement should reveal this digit information. We thus follow the standard evaluation methods in disentangled representation

learning and investigate if the learned latent variables indeed contain the digit information via latent traversal, and Figure 6 confirms that our learned latent variables capture the underlying digit structure, demonstrating physical interpretability through microstructure discovery. In the third setting of unsupervised learning, without ground truth labels, we evaluate interpretability through latent traversal analysis. Figure 5 demonstrates that our method discovers meaningful physical factors: border rotation between segments, relative fiber orientation between segments, and fiber orientation of individual segments—all physically meaningful properties for understanding the microstructure mechanism.

**Physical insights across different PDE types.** Our method is designed for parametric PDE families—scenarios where underlying physics follows the same governing equations with varying parameters. This represents a very common and scientifically important class of problems in computational physics and engineering. While our current framework focuses on parametric variation within PDE families, the core architectural principles can extend to different PDE types. For example, in multi-physics systems, our approach could handle different but related physics (e.g., heat conduction vs. diffusion-reaction) by learning shared latent factors capturing common physical principles while using task-specific decoders. Extending to heterogeneous PDE types represents an exciting research frontier that would require modular architectures for different equation types, shared representation learning across physics domains, and physics-informed constraints for cross-domain transfer.

**Accuracy-interpretability trade-off**. As discussed, MetaNO represents the forward prediction upper-bound for Disentango. The accuracy gap between the two is primarily due to insufficient latent dimensions relative to data complexity. Taking experiment 3 for instance, the data generation process may involve more than 30 true generative factors due to numerical solver noise and system complexity. Table 3 clearly shows the error decreases from 25.18% to 5.28% as we increase latent dimensions from 2 to 30—a trend that would continue with higher dimensions. We stopped at 30 dimensions because: (1) 5.28% error is reasonably accurate for practical use, and (2) limited computational resource. In practice, users should gradually increase latent dimensions until convergence to MetaNO's performance, ensuring the model capacity matches the true generative factors. DisentangO prioritizes joint modeling with interpretability over pure forward accuracy. For applications requiring only forward prediction, specialized NOs remain more accurate. However, for scientific discovery where understanding mechanisms is essential, the slightly degraded accuracy with full interpretability represents a favorable trade-off.

## C  DATA GENERATION

### C.1  EXPERIMENT 1 - SUPERVISED FORWARD AND INVERSE PDE LEARNING

we consider the constitutive modeling of anisotropic fiber-reinforced hyperelastic materials governed by the Holzapfel–Gasser–Ogden (HGO) model, whose strain energy density function can be written as:

$$\eta = \frac{E}{4(1+\nu)}(\bar{I}_1 - 2) - \frac{E}{2(1+\nu)}\ln J$$

$$+ \frac{k_1}{2k_2}(e^{k_2\langle S(\alpha)\rangle^2} + e^{k_2\langle S(-\alpha)\rangle^2} - 2) + \frac{E}{6(1-2\nu)}(\frac{J^2-1}{2} - \ln J)\,, \tag{C.1}$$

where $\langle\cdot\rangle$ indicates the Macaulay bracket, $\alpha$, $k_1$ and $k_2$ are the fiber angle, modulus and exponential coefficient, respectively, $E$ denotes the Young's modulus of the matrix, $\nu$ is Poisson's ratio, and $S(\alpha)$ describes the fiber strain of the two fiber groups, $S(\alpha) = \frac{\bar{I}_4(\alpha)-1+|\bar{I}_4(\alpha)-1|}{2}$. $\bar{I}_i$ is the $i^{\text{th}}$ invariant of the right Cauchy-Green tensor $\boldsymbol{C}$, $\bar{I}_1 = tr(\boldsymbol{C})$ and $\bar{I}_4 = \boldsymbol{n}^T(\alpha)\boldsymbol{C}\boldsymbol{n}(\alpha)$, with $\boldsymbol{n}(\alpha) = [\cos(\alpha), \sin(\alpha)]^T$. In this context, the data generation process is controlled by sampling the material set $\{E, \nu, k_1, k_2, \alpha\}$, and the latent factors can be learned consequently in a supervised fashion. The physical parameters are sampled from $\frac{k_1}{k_2} \sim \mathcal{U}[90, 100]$ ,$k_2 \sim \mathcal{U}[0.001, 0.1]$, $E \sim \mathcal{U}[0.5001, 0.6001]$, $\nu \sim \mathcal{U}[0.2, 0.3]$, and $\alpha \sim \mathcal{U}[\pi/10, \pi/2]$. To generate the high-fidelity (ground-truth) dataset, we sample 220 material sets, which are split into 200/10/10 for training/validation/test, respectively. For each material set, we sample 50 different vertical traction conditions $T_y(\boldsymbol{x})$ on the top edge from a random field, following the algorithm in Lang & Potthoff (2011b); Yin et al. (2022). $T_y(\boldsymbol{x})$ is taken as the restriction of a 2D random field, $\phi(\boldsymbol{x}) = \mathcal{F}^{-1}(\gamma^{1/2}\mathcal{F}(\Gamma))(\boldsymbol{x})$, on the top edge. Here, $\Gamma(\boldsymbol{x})$ is a Gaussian white noise random field on $\mathbb{R}^2$, $\gamma = (w_1^2 + w_2^2)^{-\frac{2}{4}}$ represents a

correlation function, and $w_1$, $w_2$ are the wave numbers on $x$ and $y$ directions, respectively. Then, for each sampled traction loading, we solve the displacement field on the entire domain by minimizing the potential energy using the finite element method implemented in FEniCS (Alnæs et al., 2015). Sample data of the obtained dataset is illustrated in Figure 9. In this setting, the model takes as input the padded traction field and learns to predict the resulting displacement field.

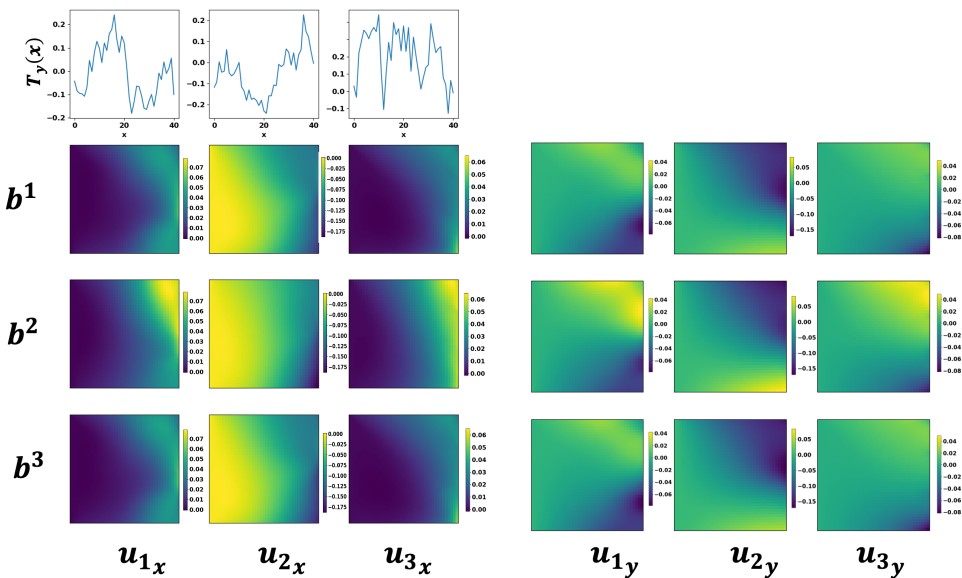

Figure 9: Illustration of the HGO data, loading and displacement pairs of three materials $b^\eta$, $\eta = 1, 2, 3$. Top: three instances of different loadings $T_y(x)$, which corresponds to different $f_i$. Bottom: corresponding displacement solutions $u_i^\eta$, illustrating the impacts of system ($b^\eta$) variability in solution operators.

## C.2 EXPERIMENT 2 - MECHANICAL MNIST BENCHMARK

Mechanical MNIST is a benchmark dataset of heterogeneous material undergoing large deformation, modeled by the Neo-Hookean material with a varying modulus converted from the MNIST bitmap images (Lejeune, 2020). It contains 70,000 heterogeneous material specimens, and each specimen is governed by the Neo-Hookean material with a varying modulus converted from the MNIST bitmap images. We illustrate samples from the MMNIST dataset in Figure 10, including the underlying microstructure, two randomly picked loading fields, and the corresponding displacement fields.

## C.3 EXPERIMENT 3 - UNSUPERVISED HETEROGENEOUS MATERIAL LEARNING

We generate two sets of datasets in this case, varying the material microstructure in the following two ways.

**Varying fiber orientation distribution.** We generate the samples by controlling the predefined parameters of Gaussian Random Field (GRF). With the GRF sharpened by the thresholding, the values to binary field represent two distinct fiber orientations. The binary field is smoothed using a windowed convolution. To address boundary conditions, the matrix is padded with replicated edge values, ensuring that the convolution works uniformly across the entire grid. After the smoothing process, the padded sections are removed, and the remaining field is used to construct the fiber field. We use two fixed fiber angles $\frac{\pi}{3}$, $\frac{2\pi}{3}$ for the corresponding binary field. We generate 300 material sets, each with 500 loading/displacement pairs, and divide these into training, validation, and test sets in a 200/50/50 split. Exemplar samples from this dataset are illustrated in Figure 11. These samples demonstrate the variability of $b$, which is critical for the latent variable identifiability in unsupervised learning settings, as proved in Theorem 2.

**Varying fiber orientation magnitude and segmentation line rotation.** Instead of controlling the fiber orientation angles on the binary field as two constant values, we sample the orientation distri-

(a) MMNIST material microstructure

(b) Loading #38 and the corresponding displacement field

(c) Loading #123 and the corresponding displacement field

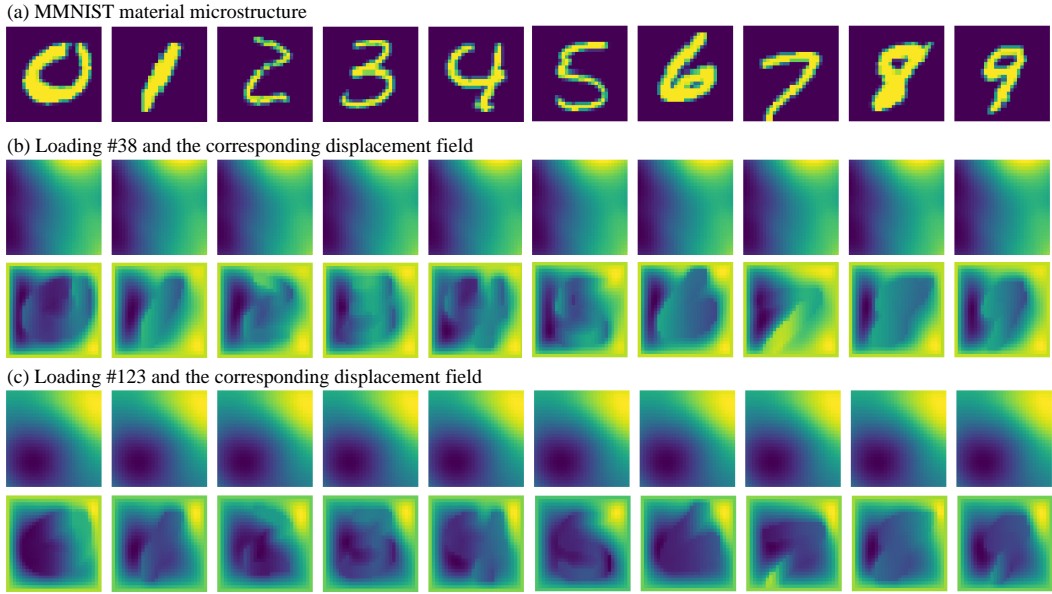

Figure 10: Illustration of exemplar MMNIST samples in the semi-supervised scenario. (a): material parameter field corresponding to different $b^\eta$. (b): displacement fields (second row) $u_{38}^\eta$ corresponding to the same loading field (first row) $f_{38}$. (c): displacement fields (second row) $u_{138}^\eta$ corresponding to the same loading field (first row) $f_{138}$.

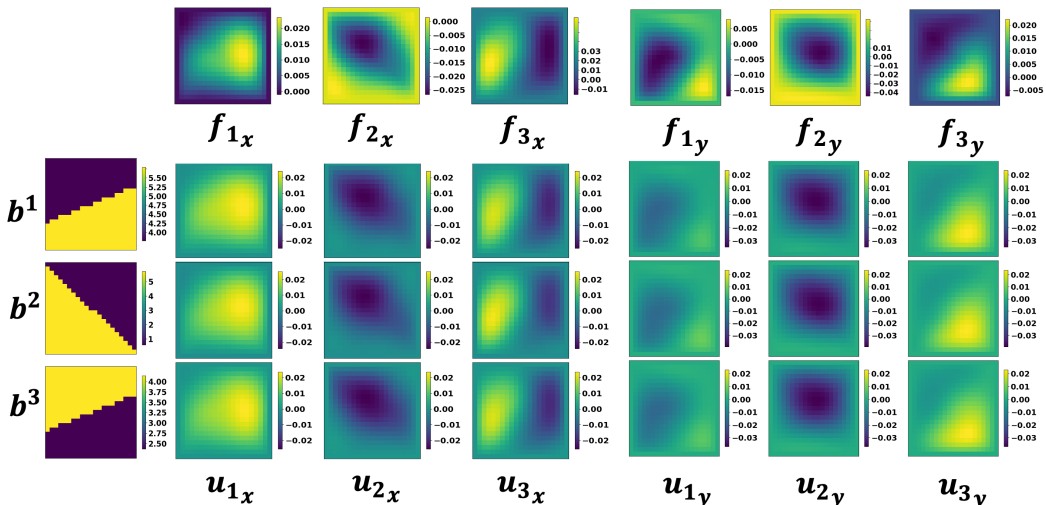

Figure 11: Illustration of fiber orientation magnitude and segmentation line rotation. Upper: three different loading instances of $f_i$. Bottom left: three different microstructure instances of $b^\eta$. Bottom right: corresponding solution fields $u_i^\eta$.

bution consisting of two segments with orientations $\alpha_1$ and $\alpha_2$ on each side, respectively, separated by a line passing through the center. The values of $\alpha_1$ and $\alpha_2$ are independently sampled from a uniform distribution over $[0, 2\pi]$, and the centerline's rotation is sampled from $[0, \pi]$. We generate 300 material sets, each with 500 loading/displacement pairs, and divide these into training, validation, and test sets in a 200/50/50 split.

**Loading and displacement pairs for one microstructure.** After generating the specimens with varying fiber orientations, we feed the $b^\eta(x)$ as the $\alpha(x)$ to the HGO model, and keep the material property set $\{E, \nu, k_1, k_2\}$ as constant. For each microstructure, we randomly sample loading and displacement pairs for each microstructure from the previous step.

The loading in this example is taken as the body load, $\boldsymbol{f}(\boldsymbol{x})$. Each instance is generated as the restriction of a 2D random field, $\phi(\boldsymbol{x}) = \mathcal{F}^{-1}(\gamma^{1/2} \mathcal{F}(\Gamma))(\boldsymbol{x})$. Here, $\Gamma(\boldsymbol{x})$ is a Gaussian white noise random field on $\mathbb{R}^2$, $\gamma = (w_1^2 + w_2^2)^{-\frac{5}{4}}$ represents a correlation function, $w_1$ and $w_2$ are the wave numbers on $x$ and $y$ directions, respectively, and $\mathcal{F}$ and $\mathcal{F}^{-1}$ denote the Fourier transform and its inverse, respectively. This random field is anticipated to have a zero mean and covariance operator $C = (-\Delta)^{-2.5}$, with $\Delta$ being the Laplacian with periodic boundary conditions on $[0, 2]^2$, and we then further restrict it to $\Omega$. For the detailed implementation of Gaussian random field sample generation, we refer interested readers to Lang & Potthoff (2011a). Then, for each sampled loading field $\boldsymbol{f}_i(\boldsymbol{x})$ and microstructure field $b^\eta(\boldsymbol{x})$, we solve for the displacement field $\boldsymbol{u}_i^\eta(\boldsymbol{x})$ on the entire domain.

### C.4 EXPERIMENT 4 - SYNTHETIC DATASET FOR IDENTIFIABILITY DEMONSTRATION

We demonstrate using experiment 4 that the disentangled latent factors from DisentangO correspond to the true generative factors. We generate synthetic data following the process in equation C.2. We work with latent variables $\boldsymbol{z}$ of dimension 2 and sample from $\boldsymbol{z} \sim \mathcal{N}(\mu_{\boldsymbol{b}}, \sigma_{\boldsymbol{b}}^2 \mathbf{I}) \in \mathbb{R}^2$ where for each microstructure $\boldsymbol{b}$, we sample $\mu_{\boldsymbol{b}} \sim \mathcal{U}[-4, 4]$ and $\sigma_{\boldsymbol{b}} \sim \mathcal{U}[1, 10]$ and $f \sim \mathcal{U}[0, 1] \in \mathbb{R}^3$, $M \sim \mathcal{U}[0, 1] \in \mathbb{R}^{2 \times 2}$, $A = [I, 2I, 3I, 4I]^T$, $M \in \mathbb{R}^{8 \times 2}$, $b \sim \mathcal{U}[0, 1] \in \mathbb{R}^8$, $W \sim \mathcal{U}[0, 1] \in \mathbb{R}^{4 \times 1}$, $l \sim \mathcal{U}[0, 1] \in \mathbb{R}$,

$$\theta = Az \,, \tag{C.2}$$

$$u = W\sigma(\theta_1 f + \theta_2) + l \,. \tag{C.3}$$

We generate 700 data pairs of $(z, \theta)$ corresponding to 700 tasks, and for each task, we generate 200 samples of $(f, u)$ pairs.

| $d$ | 3 | 5 | 7 | 9 |
|---|---|---|---|---|
| MCC | 0.7836 | 0.8013 | 0.8237 | 0.9121 |
| $R^2$ | 0.5673 | 0.6026 | 0.6473 | 0.8242 |
| Rank($w$) | 3 | 4 | 4 | 4 |

Table 5: The MCC and $R^2$ scores to evaluate identifiability on the synthetic dataset in experiment 4.

We conduct experiments on the synthetic dataset and split the tasks into $d/100/500$ for training, validation and test, respectively, with $d$ chosen from the set of $(3, 5, 7, 9)$ to investigate the effect of the number of training tasks on the identifiability. In particular, we measure the component-wise identifiability of the latent variables $\boldsymbol{z}$ by computing the Mean Correlation Coefficient (MCC) and the coefficient of determination $R^2$ scores on the test dataset, and report the results in Table 5, where we observe a monotonic growth on both MCC and $R^2$ with the increase of the number of training tasks $d$. Figure 12 illustrates the alignment on the test dataset between the true generative factors $z$ and the disentangled factors $\hat{z}$ from DisentangO in the case of $d = 9$. To verify Assumption 4, we calculate the rank of the matrix $w(\boldsymbol{z}, \boldsymbol{b})$. Specifically, for each $\theta$, with the VAE encoder, we calculate the $\mu_{\boldsymbol{b}}$ and $\sigma_{\boldsymbol{b}}$, sample $z$, and calculate the corresponding $\dfrac{\partial q_i(z_i, \boldsymbol{b}^j)}{\partial z_i}$ and $\dfrac{\partial^2 q_i(z_i, \boldsymbol{b}^j)}{\partial z_i^2}$, where $i = 1, \cdots d_z, j = 1 \cdots d$.

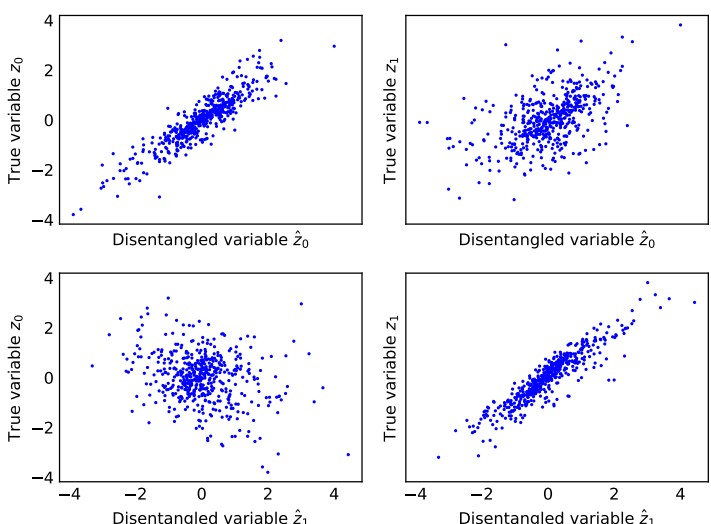

Figure 12: The scatter plot of the true generative variables $z$ and the disentangled latent variables $\hat{z}$ from the synthetic dataset with $d = 9$ in experiment 4.

