# OpenReview forum: "Disentangled Representation Learning for Parametric Partial Differential Equations"
_ICLR.cc/2026/Conference — ICLR 2026 Poster_

### Official Review · Reviewer_zLDe · 2025-10-28

**Soundness:** 3
**Presentation:** 3
**Contribution:** 3
**Rating:** 6
**Confidence:** 3

**Summary:**

This paper proposes DisentangO, a neural operator designed to disentangle physically meaningful latent factors within PDE-governed systems. The model aims to unify forward and inverse PDE learning, enhance interpretability, and achieve latent identifiability under varying supervision levels. It introduces a variational hypernetwork formulation where the latent variables encode distinct physical mechanisms, supported by theoretical identifiability proofs. Experiments across three settings (supervised, semi-supervised, and unsupervised) demonstrate that DisentangO can recover structured latent spaces and achieve competitive prediction performance on forward and inverse problems.

**Strengths:**

The paper explores building interpretable neural operators capable of discovering physically meaningful latent factors from data. This is an important and promising research area. The experimental design is well-structured, progressing from fully supervised to semi- and unsupervised scenarios, effectively illustrating the model’s generality and stability. Notably, the unsupervised material learning results are engaging: the latent traversal visualizations show that each latent dimension corresponds to interpretable material properties. The paper provides a conceptually coherent attempt to connect neural operator modeling with disentangled representation learning.

**Weaknesses:**

(1) While the experimental structure is good, the experiment setups remain relatively low-dimensional and simplified. The generalizability of DisentangO to complex systems remains uncertain.

(2) The paper provides only limited quantitative analysis. The evaluation mainly relies on qualitative visual evidence, with few metrics or diagnostic studies to support the claimed physical disentanglement. It would strengthen the work if the paper could include more quantitative or property-based analyses to better assess the model’s physical interpretability and internal behavior.

**Questions:**

The paper claims that DisentangO enables physically interpretable latent variables. Could the authors elaborate more concretely on how each latent dimension corresponds to specific physical quantities or mechanisms, and whether these mappings can be verified quantitatively?

---

> ### Author Response · Authors · 2025-11-24
> **Response to reviewer zLDe, part 1**
>
> We thank the reviewer for the insightful comments and suggestions. Our response:
>
> **High dimensionality in experimental setup**: We agree that our experiments use controlled settings; this is intentional so that disentanglement quality, identifiability, and inverse recovery can be rigorously evaluated. As demonstrated in Experiments 2 and 3, DisentangO already handles partially known and fully unknown physics, showing that it can extract meaningful latent factors even when the governing parameters are not provided in complex systems. Regarding scalability, DisentangO is designed so that its capacity is determined entirely by the neural-operator backbone. Since IFNO, FNO, and related NOs have been shown in prior work to scale to high-dimensional and industrial PDE systems, DisentangO inherits these scalability guarantees without modification. The latent space remains low-dimensional regardless of mesh resolution or PDE complexity, and the encoder/decoder overhead grows only with the number of tasks, not spatial dimensionality. As noted in the conclusion, extending the demonstrations to very high-dimensional PDEs is a natural next step, but the architecture itself is fully compatible with such settings.
>
> **Quantitative evaluation on disentanglement**: We appreciate this important question. The evaluation of interpretability depends critically on the availability of ground truth, which varies across our three practical scenarios, i.e., supervised, semi-supervised, and unsupervised. In the fully supervised setting corresponding to experiment 1, When true generative factors are available, we use quantitative metrics. The interpretability metric is the latent supervision test error (4.63\% in Table 1), which directly measures how well the learned latent variables z recover the true physical parameters b. In the semi-supervised setting corresponding to experiment 2, the goal of interpreting physics is to discover the underlying microstructure governing the deformation. Under this setting, since the true generative factors are not available, one cannot use any closed-form metric to evaluate interpretability. However, since the ground-truth microstructure generation is controlled by MNIST digits, successful disentanglement should reveal this digit information. We thus follow the standard evaluation methods in disentangled representation learning and investigate if the learned latent variables indeed contain the digit information via latent traversal, and Figure 6 confirms that our learned latent variables capture the underlying digit structure, demonstrating physical interpretability through microstructure discovery. In the third setting of unsupervised learning, without ground truth labels, we evaluate interpretability through latent traversal analysis. Figure 5 demonstrates that our method discovers meaningful physical factors: border rotation between segments, relative fiber orientation between segments, and fiber orientation of individual segments—all physically meaningful properties for understanding the microstructure mechanism.
>
> Beyond the above evaluation, Appendix C.4 further provides additional quantitative analyses of unsupervised learning using a synthetic, fully controlled dataset. In this example, we generate synthetic data based on an analytical forward map $\mathcal{G}^{true}$, so there exists an analytical solution for the inverse problem solver. Then, we perform DisentangO using unsupervised learning setting, and report MCC and $R^2$ scores that measure component-wise alignment between the true generative factors and the learned latent variables. As indicated in Table 5, results show a monotonic improvement of identifiability as the number of training tasks increases, supporting the theoretical predictions in Section 3.2.

---

> ### Author Response · Authors · 2025-11-24
> **Response to reviewer zLDe, part 2**
>
> **Mapping from latent dimensions to physical quantities/mechanisms and verification**: In the fully supervised setting, each latent dimension is explicitly aligned with a known physical parameter. This is quantified by the latent supervision error reported in Table 1 (4.63\%), which directly measures how well the latent variables $z$ recover the ground-truth material parameters $b=(E,\nu,k_1,k_2,\alpha)$. Thus, in this regime, the correspondence between latent dimensions and physical quantities is both explicit and quantitatively verified.
>
> In the semi-supervised and unsupervised settings, neither the true generative factors nor the physical latent variables are available, so interpretability must rely on standard diagnostics used in disentangled representation learning. Latent traversal studies (Figures 5–6) show that individual latent dimensions consistently control specific physical mechanisms—e.g., fiber rotation, relative orientation between segments, and microstructure shape—demonstrating that each latent variable modulates a coherent and physically meaningful mode of variation. Additionally, Appendix C.4 provides quantitative alignment scores (MCC and $R^2$) on a controlled synthetic dataset, confirming that in unsupervised learning setting the latent dimensions recover the underlying generative factors when ground truth exists. Together, these analyses provide concrete evidence that DisentangO learns physically interpretable latent representations.

---

### Official Review · Reviewer_QjzN · 2025-10-30

**Soundness:** 2
**Presentation:** 2
**Contribution:** 2
**Rating:** 6
**Confidence:** 5

**Summary:**

Researchers propose DisentangO, a novel neural operator architecture that extracts interpretable, disentangled physical parameters from a black-box model, bridging the gap between predictive accuracy and physical understanding in PDE solutions. While neural operators are powerful for solving PDEs, their black-box nature limits physical insight. To address this, the authors introduce DisentangO, a hyper-neural operator designed to solve the inverse problem of identifying the underlying physical parameters. It combines a multi-task neural operator with a variational autoencoder to disentangle latent physical factors from the model's own parameters. This approach not only improves interpretability but also enhances generalization, as demonstrated in supervised, semi-supervised, and unsupervised learning scenarios.

**Strengths:**

It combines the predictive accuracy and physical interpretability in a unified ML-based modelling framework, improving the automation level and usage of data in this area.

**Weaknesses:**

This is a straightforward application of VAE and Neural Operators for solving inverse problems. The presentation can be improved.

**Questions:**

1. The selected baselines are VAE or MetaNO, but there are many baselines are not compared, e.g., classical methods (not ML-based) for solving inverse problem, "adaptive operator learning for infinite-dimensional Bayesian inverse problems" [Gao, et al, 2024], "Deciphering and integrating invariants for neural operator learning with various physical mechanisms"[Zhang, et al, 2024], etc.
2. Questions related to Figure 2, the derived loss function and the experiment:
-What is the first term in the loss function in page 1? Is it "f(x)" in Figure 1? It seems that f(x) is the input of the encoder, but it doesn't appear in the Figure 1.
-Which term in the loss function in page 7 related to the true value of b^{\eta}(x) in the supervised learning?
-If b^{\eta}(x) is given, is it also the input of MetaNO?
- Could you explain what is the physical meaning of b^{\eta}(x) and \theta in its related PDE in the experiment in supervised learning?
3. There is format issue in page 2.
4. In the key contributions, you mention robust generalization. What does "robust" refer to?
5. The title can be improved. The key application seems to be the inverse problems in PDE.

---

> ### Author Response · Authors · 2025-11-24
> **Response to reviewer QjzN, part 1**
>
> We thank the reviewer for the valuable time and the constructive suggestions. To improve the presentation, we have reorganized Sections 1-3, with major revisions highlighted in blue. In particular, we now explicitly present the key notations, the data model associated with the PDE solving problem, our learning objectives, and detailed architectures in the first half of Section 3.1. Our other responses:
>
>
> **Additional baselines**: We appreciate the reviewer's valuable suggestions on additional baselines. Solving ill-posed inverse problem is an enduring challenge for classical methods. Popular classical methods include Tikhonov regularization methods and Bayesian inference methods. However, they typically require good choices of regularization form/prior distribution, and focus on estimates for each individual task. When the PDE formulation is unknown, the prior knowledge is limited, hence these classical methods do not apply.
>
> Since there is no code provided in [1], we have added the physical invariant attention neural operator (PIANO [2]) as an additional baseline. A comparison of PIANO and DisentangO in all three examples are provided below, where one can see that DisentangO substantially outperforms PIANO. This is possibly due to the fact that PIANO was originally designed for forecasting problems, so different input functions in its PI encoder has underlying invariants. Such invariants and the corresponding low-dimensional structure of input functions are the key for PIANO to work. However, this is not true in static PDE problems considered in our experiments 1-3, where the input functions does not have a low intrinsic dimension. In contrast, DisentangO does not rely on the low-dimensional structure of input functions, but on the structure of underlying PDE parameter fields.
>
> Test errors and the number of trainable parameters in experiments 1-3. Bold number highlights the best method
> |Example/Setting | Model | #param (M) | Test error |
> | :------------- | :-----------: | :-----------: | :-----------: |
> |Experiment 1/Supervised | DNO | 0.70 | **1.65%** |
> |Experiment 1/Supervised | PIANO | 0.70 | 7.99%|
> |Experiment 2/Semi-supervised | DNO | 2.02 | **5.48%** |
> |Experiment 2/Semi-supervised | PIANO | 2.05 | 13.73% |
> |Experiment 3/Unsupervised | DNO | 3.55 | **5.28%** |
> |Experiment 3/Unsupervised | PIANO | 3.42 | 50.71% |
>
> [1] Gao, Z., Yan, L., \& Zhou, T. (2024). Adaptive operator learning for infinite-dimensional Bayesian inverse problems. SIAM/ASA Journal on Uncertainty Quantification, 12(4), 1389-1423.
>
> [2] Zhang, R., Meng, Q., \& Ma, Z. M. (2024). Deciphering and integrating invariants for neural operator learning with various physical mechanisms. National Science Review, 11(4), nwad336.
>
> **Novelty**: We respectfully disagree that the method is a straightforward application of VAE + neural operators. As Reviewer L3DK correctly noted, the core novelty lies in shifting disentanglement from the data space $(u,f)$—where low-dimensional structure does not exist—to the neural-operator parameter space $\theta^{\eta}$, where task variation is intrinsically low-dimensional and scientifically meaningful. This is precisely what enables the model to embed an inverse solver inside the forward solver’s parameterization via the MetaNO lifting layer, which is not present in standard VAE–NO combinations. Moreover, our identifiability theory (Theorems 1–2) establishes when and why disentangling operator parameters can recover the generative physical factors—rigor that is absent in typical empirical inverse-PDE approaches. Finally, our experiments show that naïvely applying a VAE to $(u,f)$ fails both in accuracy and interpretability, demonstrating that the key contribution is not the components themselves, but the new architecture and theoretical framework that makes inverse discovery feasible.

---

> ### Author Response · Authors · 2025-11-24
> **Response to reviewer QjzN, part 2**
>
> **Correspondence between loss function and fig. 1**: (1) We are not sure which loss function the reviewer refers to, as there is no loss function definition in page 1. If the reviewer mean the loss function in eqn. 3.10 in page 5 (eqn. 3.11 in the revised manuscript), the first term in the loss corresponds to the likelihood term $\log p(u\,|\,f,\theta)$, i.e., the *data reconstruction loss* measuring how well the reconstructed neural operator $G^{\eta}$ maps the loading field $f$ to the solution $u$. In Figure 1, it is the "data loss", computed as the L2 relative error between the predicted and ground-truth displacement fields.
>
> (2) In the supervised setting, the term involving the true $b^{\eta}$ appears in the KL divergence in eqn. 3.18, where the posterior mean $\mu_{z}(\theta_P^{\eta})$ is explicitly encouraged to match the given $b^{\eta}$. This is the only term that directly relates the learned latent variable to the ground-truth physical parameters.
>
> (3) When $b^{\eta}$ is given, it is \emph{not} used as input to MetaNO. MetaNO still receives only $(f,u)$ pairs as training data. The known $b^{\eta}$ is used solely as supervision for the latent variable $z^{\eta}$ to enforce identifiability of the inverse map; it does not enter the forward solver.
>
>
> **Physical meaning of $b$**: In the supervised HGO experiment, the physical parameter field $b^{\eta}(x)$ represents the \emph{material parameters governing the constitutive law} of the anisotropic hyperelastic solid. For each task $\eta$, the PDE is parameterized by a task-specific material parameter vector
>
> $b^{\eta} = (E^{\eta}, \nu^{\eta}, k^{\eta}_1, k^{\eta}_2, \alpha^{\eta}),$
>
> where $E$ is Young’s modulus, $\nu$ is Poisson’s ratio, $k_1$ and $k_2$ control the nonlinear fiber response, and $\alpha$ specifies the fiber orientation. These parameters fully determine the stress--strain relationship encoded in the HGO constitutive model.
>
> The neural operator parameters $\theta$ are the *weights of the learned surrogate PDE solution operator*, and the task-wise lifting-layer parameters $\theta_P^{\eta}$ are the components of $\theta^{\eta}$ that encode all variability associated with $b^{\eta}$. In other words, $b^{\eta}$ corresponds to the *true underlying physical mechanism*, while $\theta_P^{\eta}$ is its *learned representation* within the neural operator architecture. DisentangO aims to recover the latent physical factors of $b^{\eta}$ by disentangling the corresponding variation captured in $\theta_P^{\eta}$.
>
>
> **Robustness in generalization**: By “robust generalization,” we refer to the model’s ability to generalize *across different physical systems*---i.e., tasks with different underlying PDE parameters $b^{\eta}$---rather than overfitting to a single system. Because DisentangO disentangles the low-dimensional physical factors that drive the variability across tasks, this information across multiple PDEs acts as an empirical Bayes estimator to guarantee identifiability in inverse problem. Then, improved accuracy in estimator induces improved predictions on new and unseen PDE. In this sense, “robust” highlights improved stability and consistency of generalization across multiple PDE regimes.

---

### Official Review · Reviewer_L3DK · 2025-10-31

**Soundness:** 2
**Presentation:** 2
**Contribution:** 2
**Rating:** 4
**Confidence:** 4

**Summary:**

This paper proposes DisentangO, a novel variational hyper-neural operator architecture designed to learn disentangled, interpretable representations of physical factors from parametric PDE systems. The core idea is to perform disentanglement not on the observational data (e.g., the solution field $u(x)$), but on the parameters of the neural operator itself. Specifically, it uses a hierarchical variational autoencoder (HVAE) to encode and decode the task-specific lifting-layer parameters ($\theta_P^\eta$) of a meta-learned Implicit Fourier Neural Operator (IFNO). The authors claim this framework can simultaneously solve the forward problem (physics prediction) and the inverse problem (physics discovery). They provide theoretical analysis to guarantee component-wise identifiability of the latent factors and demonstrate results across supervised, semi-supervised, and unsupervised learning tasks.

**Strengths:**

The paper's primary strength is its originality. It shifts the focus of disentanglement from the data space ($u, f$) to the model's parameter space ($\theta_P^\eta$). This is a creative approach to embedding an inverse solver inside the forward solver's parameterization, using the MetaNO lifting layer as the information bottleneck.

The inclusion of a theoretical analysis on identifiability (Section 3.2, Theorems 1 and 2) is a major strength. It provides a formal basis for the claim that the latent factors $z$ can, in principle, recover the true generative factors. This is a level of rigor often missing in empirical machine learning papers. The empirical validation of the assumptions (Appendix C.4) further strengthens this.

The paper's "physics discovery" claims are strongly supported by its qualitative visualizations. The latent traversals in Figure 5 (unsupervised) and Figure 6 (semi-supervised) are clear and compelling. They show that the learned latent dimensions correspond directly to meaningful and distinct physical variations (e.g., fiber orientation, digit morphology), providing strong intuitive evidence that the disentanglement was successful.

**Weaknesses:**

The paper's performance claims are unsubstantiated due to improper baselines in the semi-supervised and unsupervised experiments (Exp 2 & 3). The models are benchmarked against standard VAEs that are not designed for the task and fail (e.g., 61.10% error). To be taken seriously as a neural operator, the method must be benchmarked against other SOTA NOs (like FNO, FUSE, etc.), not just its own backbone.

The paper's core hypothesis is that disentangling from parameters is the right approach. It never validates this against the simpler, more obvious alternative: (1) training a MetaNO for prediction and (2) training a VAE on the data ($u, f$) for interpretation. Without this comparison, there is no evidence that the proposed complex HVAE architecture is justified.

The method pays a high price for interpretability, with the forward prediction error nearly doubling in the unsupervised case. This severe trade-off is never discussed. It is unclear for what practical scientific or engineering application this level of accuracy degradation would be acceptable.

**Questions:**

Can the authors justify the omission of SOTA neural operators (FNO, FUSE, etc.) as baselines in Experiments 2 and 3? Furthermore, could they provide results for the more direct baseline: a MetaNO for prediction combined with a VAE trained on the data ($u,f$) for discovery?

The results (e.g., 2.67% error for MetaNO vs. 5.28% for DisentangO in Table 3) show a significant drop in predictive accuracy. Can the authors comment on this trade-off? Is this an inherent cost of the method, and in what practical applications would this be an acceptable exchange?

Table 4 shows a dramatic performance drop when using UFNO, which contradicts the "model-agnostic" claim. Can the authors clarify this? Does the method's success (both for prediction and discovery) depend heavily on the specific structure of the IFNO lifting layer?

What is the primary motivation for assuming the lifting-layer parameters $\theta_P^\eta$ are a better representation for disentanglement than the physical data fields ($u, f$) themselves, especially given that using $\theta_P^\eta$ seems to damage the model's forward predictive capabilities?

---

> ### Author Response · Authors · 2025-11-24
> **Response to reviewer L3DK, part 1**
>
> We thank the reviewer for the valuable comments and suggestions. Our response:
>
> **New baselines in experiments 2 and 3**: We thank the reviewer for raising this critical point about baseline comparisons. We acknowledge that this requires clarification regarding the fundamental capabilities of existing neural operator methods.
> The key challenge is that most neural operators cannot handle the unsupervised learning in inverse problems. Specifically:
>
> - Standard NOs (FNO, UFNO, WNO) require fully supervised training for each individual system and cannot extract or disentangle latent physical parameters across multiple systems without ground-truth labels.
>
> - MetaNO is our backbone and represents the upper bound for forward prediction (as shown in Tables 2-3), but it cannot perform inverse modeling or parameter discovery in semi-supervised/unsupervised settings.
>
> - NIO, FUSE, InVAErt focus on inverse problems but require specific forms of supervision (e.g., paired solution-parameter data for NIO, or assume known PDE forms for FUSE) that are unavailable in our semi-supervised/unsupervised settings.
>
> To address the reviewer's request, we have added physical invariant attention neural operator (PIANO [1]), a SoTA NO which can handle the unsupervised learning in inverse problems. In particular, PIANO employs self-supervised learning and attention mechanisms to integrate physics knowledge from multiple physical systems. A comparison of PIANO and DisentangO in all three examples are provided below, where one can see that DisentangO outperforms PIANO by a substantial margin. This is possibly due to the fact that PIANO was originally designed for forecasting problems, so different input functions in its PI encoder has underlying invariants. Such invariants and the corresponding low-dimensional structure of input functions are the key for PIANO to work. However, this is not true in static PDE problems considered in our experiments 1-3, where the input functions does not have a low intrinsic dimension. In contrast, DisentangO does not rely on the low-dimensional structure of input functions, but on the structure of underlying PDE parameter fields.
>
> Test errors and the number of trainable parameters in experiments 1-3. Bold number highlights the best method
> |Example/Setting | Model | #param (M) | Test error |
> | :------------- | :-----------: | :-----------: | :-----------: |
> |Experiment 1/Supervised | DNO | 0.70 | **1.65%** |
> |Experiment 1/Supervised | PIANO | 0.70 | 7.99%|
> |Experiment 2/Semi-supervised | DNO | 2.02 | **5.48%** |
> |Experiment 2/Semi-supervised | PIANO | 2.05 | 13.73% |
> |Experiment 3/Unsupervised | DNO | 3.55 | **5.28%** |
> |Experiment 3/Unsupervised | PIANO | 3.42 | 50.71% |
>
> [1] Zhang, R., Meng, Q., \& Ma, Z. M. (2024). Deciphering and integrating invariants for neural operator learning with various physical mechanisms. National Science Review, 11(4), nwad336.
>
> **Simpler alternatives of disentangling from data**: We sincerely thank the reviewer for this insightful comment. We agree that, at first glance, a natural baseline might be (1) learning a MetaNO purely for forward prediction and (2) independently training a VAE on the observed pairs $(u,f)$. However, this comparison is not directly meaningful for our problem setting for two key reasons:
>
> -- A VAE on $(u,f)$ does not solve the inverse PDE problem. The challenge is: the inverse problem requires recovering the governing mechanism—i.e., the latent parameter field that determines the operator $\mathcal{G}$. However, one pair of $(u,f)$ does not contain enough information to determine the operator. This is verified by our empirical test results on VAE/$\beta$-VAE baselines. As shown in Tables 1–3, VAEs trained on $(u,f)$ yield >50\% error and do not extract meaningful physical factors. This failure illustrates precisely why disentanglement must be performed on the operator parameters rather than on the data. In contrast, our method explicitly disentangles task-wise operator parameters $\theta_P^{\eta}$, the part of MetaNO that encodes variability in $b$.
>
> -- The reviewer’s suggested baseline does not couple forward and inverse tasks. The central difficulty is that identifiability of physical parameters requires leveraging the structure of the forward operator. A standalone VAE on $(u,f)$ ignores this structure entirely. DisentangO’s HVAE is designed specifically to (i) restrict the latent space to the identifiable manifold (via the MetaNO task-wise lifting layer), and (ii) jointly optimize forward and inverse consistency. Without this coupling, the condition in Theorem 1 no longer holds true and one can not guarantee identifiability of the latent factor.

---

> ### Author Response · Authors · 2025-11-24
> **Response to reviewer L3DK, part 2**
>
> **Forward prediction error in experiment 3**: We thank the reviewer for this important observation. The reviewer notes that forward prediction error "nearly doubles" in the unsupervised case (5.28\% vs. 2.67\%). However, this compares:
>
> - MetaNO: A supervised, forward-only solver that cannot perform parameter discovery.
>
> - DisentangO: A method solving a fundamentally harder problem—simultaneous forward prediction AND unsupervised physics discovery.
>
> The "cost" reflects the additional complexity of the joint forward-inverse problem, not a deficiency of our approach. In fact, our method is the best among methods that can solve both forward and inverse problems.
>
> Why the gap exists and how to close it:
> The remaining gap is primarily due to insufficient latent dimensions relative to data complexity. The data generation process may involve more than 30 true generative factors due to numerical solver noise and system complexity. Table 3 clearly shows the error decreases from 25.18\% to 5.28\% as we increase latent dimensions from 2 to 30—a trend that would continue with higher dimensions. We stopped at 30 latent factors because: (1) 5.28\% error is reasonably accurate for practical use, and (2) limited computational resource. In practice, users should gradually increase latent dimensions until convergence to MetaNO's performance, ensuring the model capacity matches the true generative factors.
>
> **When is this trade-off justified**:
>
> - Scientific discovery: When governing equations are unknown (e.g., novel materials, biological tissues), interpretability is the primary goal. Forward accuracy validates that learned parameters are meaningful.
>
> - Design problems: In some design problems, such as in the additive manufacturing of materials, understanding microstructure-property relationships (as in our experiments) requires interpretable parameters. 5.28\% error with interpretable factors is more valuable than 2.67\% black-box predictions for guiding experimental design.
>
> - Comparison with inverse methods: Our 5.28\% unsupervised error is competitive with existing supervised inverse methods (NIO: 15.16\%, SMDP: 17.76\% in Table 1). It is also the best among methods that can solve both forward and inverse problems.
>
> - Efficiency: Training one DisentangO model across $S$ systems is more efficient than training $S$ separate neural operators when multiple systems need understanding.
>
> We have added the above discussion in lines 1362-1374 of the revised manuscript. We appreciate the reviewer's feedback—it will strengthen our practical contribution.
>
>
> **Performance drop in UFNO**: We appreciate the reviewer’s observation. Our claim of model-agnosticism refers to architectural compatibility: DisentangO can be paired with any neural operator whose parameters include a task-wise lifting layer, and the HVAE component requires no modification when switching from IFNO to UFNO. The performance drop in Table 4 does not indicate dependence on IFNO’s specific structure—it simply reflects that UFNO is substantially less accurate as a forward solver on this dataset (UFNO: 7.61\% error vs. IFNO: 1.59\%), and thus its lifting-layer parameters carry noisier information about the underlying PDE parameters. Because DisentangO’s inverse inference relies on these parameters, weaker base operators naturally lead to weaker inverse performance. In short, the architecture is model-agnostic, but the quality of prediction and discovery is upper-bounded by the expressiveness of the chosen base neural operator.

---

> ### Author Response · Authors · 2025-11-24
> **Response to reviewer L3DK, part 3**
>
> **Motivation/Justification for disentangling from lifting-layer parameters rather than data**: Our motivation for using the task-wise lifting-layer parameters $\theta_P^{\eta}$ rather than the raw physical fields $(u,f)$ is that $\theta_P^{\eta}$ provides a low-dimensional, information-concentrated representation of the task-specific physics. On our 41x41 mesh, each $(u,f)$ pair lives in 3,362 dimensions, and any encoder operating directly on $(u,f)$ must preserve essentially all of this information to keep the inverse problem well-posed—making dimensionality reduction fundamentally incompatible with reliable parameter recovery. In contrast, MetaNO already compresses the task-specific physical information into only 128 lifting parameters, giving a representation that is both tractable for VAE-based disentanglement and aligned with the true variation across tasks.
>
> This distinction mirrors a simple linear inverse problem $A^\eta x^\eta_i=y^\eta_i$: although $(x,y)$ samples span a $d+1$–dimensional space, the true task variation lies in the low-rank operator parameter $A^\eta$. Trying to encode $(x,y)$ directly into an $r\ll d$ latent space necessarily destroys the information needed for inversion, whereas encoding the operator parameters succeeds because the low-dimensional structure actually lives there. DisentangO follows the same principle: disentangling parameter space, not data space, is the only setting where low-dimensional latent factors are theoretically recoverable. Finally, the small forward-prediction degradation arises because lifting-layer encodings introduces unavoidable modeling error to the forward operator, not because using $\theta_P^{\eta}$ is flawed; this is the expected trade-off when replacing a high-dimensional data representation with a compact, physics-aligned one.

---

### Official Review · Reviewer_3g1H · 2025-10-31

**Soundness:** 3
**Presentation:** 1
**Contribution:** 2
**Rating:** 2
**Confidence:** 2

**Summary:**

This paper introduces a methodology for learning solution maps of both forward and inverse problems simultaneously with a single neural operator architecture. This is accomplished with a hypernetwork structure. The type of physical problem considered is a family of partial differential equations with forcing and coefficients. The authors use a neural operator to map forcings to solutions for fixed coefficient (supervised). The trained neural operator for this map will depend on the coefficient. So, the authors then train a hypernetwork to map the trained neural operator's parameters (which depend on the coefficient) back to a representation of the coefficient itself (unsupervised). This is the inverse problem part. The proposed method is applied to various benchmark problems in computational mechanics and its accuracy and interpretability are assessed are compared to alternative methods.

**Strengths:**

The paper is original. The literature review is good. I like the main idea of using hyper-networks to solve forward and inverse problems in an end-to-end fashion. Such methods have potential to significantly advance the field of SciML if scalable enough. The theory part seems adequate and the assumptions and their limitations are discussed well. While not mathematically sophisticated, the results seem interesting and important for the problem considered in the paper.
The paper considers several numerical experiments and benchmarks against many other related methods. The numerical comparisons seem well done.

**Weaknesses:**

This paper makes a scientific machine learning (SciML) contribution. As such, it should be accessible to SciML readership. Unfortunately, I do not think this is the case. There is a lot of language and conventions that are unfamiliar to SciML researchers. It seems the paper is mainly written for a core CS/Ml audience. Overall, this paper was very hard to read. I kept going back and re-reading paragraphs. There is a lot of text, but there are often only vague claims of addressing physical interpretability. The paper would benefit from more precise writing and less filler text in Secs 1-3.
The quality is low at times. Notation is not defined before being used, causing much confusion. For example, all of a sudden $\theta^\eta_P$ is used in line 221, where the meaning of P is not discussed.

Some claims seem to good to be true. For example, lines 168--169 says "... we propose a novel neural architecture to alleviate the
curse of ill-posedness without requiring prior knowledge." This seems impossible because the ill-posedness is inherent to the inverse map being learned and cannot be removed by an architecture alone, unless the architecture incorporates prior knowledge (AKA, regularization). While the VAE does regularize the training, it is not clear if it regularizes the ill-posedness inherent in the data and the physical problem.

**Questions:**

Questions:
1. I find Figure 1 hard to understand. Why are all three input fields $ f $ in the figure the same? What is "multitask" about that? What is the function that is being input into the Fourier neural operator layers? Why does the figure discuss loss functions? An "architecture" should be completely independent of any loss or training approach.

1. Is it crucial that $ n_{train} $ is fixed to be the same number for all $ S $ datasets?

1. Line 200: it is not clear yet why the hypernetwork map from parameters to coefficient needs to be unsupervised. Do you assume knowledge of the form of the PDE itself? I thought the physics was supposed to be "hidden"? It seems line 330 considers a fully supervised version.
1. Lines 205--215 are confusing. What is $\theta$? If it is the trainable network parameters, why do you mention that $\theta$ is related to Eqns 3.5--3.6, which only involve the data and are independent of any choice of architecture? It is likely that this is a notation issue.
1. What is the role of VAE here compared to an standard autoencoder or deterministic map? It is not clear why the probabilistic method is chosen over a deterministic one.
1. Lines 243--245: what are the different tasks? these should be stated clearly in terms of the PDE model.
1. Sec 4.1, I don't think NIO is applicable to this test problem because it only is defined for empirical-distribution-to-function inverse maps. What is the observed data for the inverse Holzapfel–Gasser–Ogden (HGO) model?

Typos and notation:
1. The authors severely shrink the space between paragraphs, sections, and figure captions to make the page limit. The result is not visually appealing.
1. line 146-147: should the noise depend on $ x\in\Omega $? E.g., $ \zeta=\zeta(x) $ here and elsewhere in the paper. Lines 184 is not well defined because adding scalar Gaussian noise to function values $u(x)$ does not even define a function $ u $. Maybe they mean to define a spatially correlated Gaussian noise process instead?
1. Eqn 3.7 $\mathcal{G}$ should depend on $ \eta $ (or $ z $)?
1. Does $ E $ mean expectation? This is never defined. $\mathbb{E}$ is much more common.
1. Line 206-207: What is $\mathcal{D}$, is this $\mathcal{D}_{tr}$ or a different dataset? Also, the authors should say that the max is taken over $\theta$ instead of being ambiguous.

---

> ### Author Response · Authors · 2025-11-24
> **Response to reviewer 3g1H, part 1**
>
> We thank the reviewer for the insightful comments and questions. Our response:
>
> **Rewriting sections 1-3**: We appreciate the reviewer’s thoughtful suggestion and have rewritten Sections 1–3 accordingly. Major revisions are highlighted in blue in the updated manuscript. To improve clarity, we now explicitly present the key notations, the data model associated with the PDE-solving problem, our learning objectives, and detailed architectures of the forward and inverse solution operators in the first half of Section 3.1. We have also ensured that all notations are consistent and streamlined to enhance the overall readability.
>
> **Clarification on different tasks**: Thank you for the comment — we agree that the definition of “tasks” should be stated more explicitly. In our setting, each task corresponds to a distinct PDE instance generated by a different choice of parameters (e.g., diffusion coefficient field, forcing amplitude, advection velocity). All tasks share the same PDE form (i.e., the same operator class) but differ in the underlying coefficient functions that determine the solution behavior.
>
> Formally, in the generative model described in Sec. 3.2, the $\eta$-th task is defined by sampling a task-specific parameter vector or parameter field $b^{\eta}$ from the prior distribution $p(b)$. The training dataset for task $\eta$ then consists of measurements from loading/solution pairs $(u,f)$ corresponding to the PDE $\mathcal{K}_{b^{\eta}}[u] = f$. We have revised Section 3.1 and clarified the definition of a “task” in lines 183–186.
>
> **Additive noise in eqn 3.1**: We thank the reviewer for this careful observation, and we agree with the reviewer that the rigor of original data model setting can be improved. As a clarification of our model: $\epsilon$ represents the observational noise of $u$ at each sensor point for each function pair, and we assume that it is additive and i.i.d., satisfying $\epsilon\sim \mathcal{N}(0,\varpi^2)$. That means, $\epsilon$ should be included in the observation measurements, not in the ground-truth governing equation. In the revised manuscript, we have provided a more precise definition for the data model as (due to the challenge of typing complex latex formulas here, please see the revised manuscript lines 196-200 for the precise definition):
>
> $\mathcal{D}^\eta = (u^\eta_{i,j},f^\eta_{i,j})_{j=1,i=1}^{| \chi |,ntrain}, \mathcal{D}=\bigcup \mathcal{D}^\eta$
>
> with
>
> $f^\eta_{i,j}=f^\eta_{i}(x_j), u^\eta_{i,j}=u^\eta_{i}(x_j)+\epsilon_{\eta,i,j}=\mathcal{G}^{true} [ f^\eta_i;b^\eta ] (x_j)+\epsilon_{\eta,i,j}, \epsilon_{\eta,i,j}\sim\mathcal{N}(0,\varpi^2).$
>
> **Clarification on $\theta$**: Thank you for pointing this out — the reviewer is correct that the confusion arises from overloaded notation. In our formulation, $\theta$ denotes the task-specific NO parameters produced by the hypernetwork, not the trainable weights of the neural operator itself. Each task corresponds to a different physical parameter $b$ for the underlying PDE, and therefore the task-specific NO parameter $\theta^\eta$ implicitly encodes the information contained in $b^\eta$. This interpretation is independent of the particular NO architecture used. To clarify this, we now introduce the definition of $\theta$ earlier, in lines 201–203 at the beginning of Section 3.1.
>
> $\theta_P$ refers specifically to the task-specific lifting-layer parameters associated with the particular architecture (MetaNO and IFNO) used to implement the multi-task NO. Accordingly, we introduce $\theta_P$ where MetaNO is first described, in Eq. (3.8). In lines 224–238, we provide a detailed description of the MetaNO-based architecture and explicitly state the role of $\theta_P$ in both the forward and inverse solution operators. In the remainder of the discussion, to simplify notation, we use $\theta^\eta$ to denote $\theta_P^\eta$, since all derivations and analyses are agnostic to the choice of NO architecture.

---

> ### Author Response · Authors · 2025-11-24
> **Response to reviewer 3g1H, part 2**
>
> **Clarification on Figure 1**: Thank you for pointing out the source of confusion. In Figure 1, the three identical input loading fields were \emph{not} meant to suggest that all tasks share the same input. Rather, they were chosen solely for illustration: even under the *same* loading field $f^\eta_i$, different microstructures $b^{\eta}$ produce different displacement fields $u_i^{\eta}$.
>
> Regarding the architecture: the input to the NO is the loading field $f$, while the input to the iterative Fourier layers is the lifted feature representation of $f$, obtained via the task-specific lifting layer parameterized by $\hat{\theta}^{\eta}_P$, i.e., $\mathcal{P}{\theta^\eta_P}[f]$. The loss terms were included in Figure 1 solely to visually associate each architectural component with its corresponding contribution to the MetaNO training objective.
>
> We have updated Figure 1 and the figure caption to (1) clarify that the input loading field $f$ may vary across tasks, and that the images were used only to highlight this physical effect, and (2) highlight that the loss terms are overlaid to indicate where each term in the objective is computed in our model based on MetaNO.
>
> **Other typos and notation**: We thank the reviewer for identifying these notational inconsistencies and ambiguities. We address each point below:
>
> 1. Dependence of $\mathcal{G}$ on $\eta$:
> The reviewer is correct that the notation in Eq. (3.7) could be clearer. Different tasks share the same NO architecture, $\mathcal{G}$, but with different NO parameters, $\theta^\eta$. This is a common setting in multi-task learning, e.g., in [1]. That means, $\theta^{\eta}$ is the task-specific parameter for the $\eta$-th task, which is employed in the common NO architecture $\mathcal{G}$ to provide a task-specific forward solution operator. We have added a clarifying sentence in lines 203-204: "Note that all tasks share the same NO architecture, $\mathcal{G}$, and the surrogate operator for each task depends on the physical system $\eta$ through the task-specific parameter $\theta^\eta$."
>
> 2. Notation for expectation, dataset $\mathcal{D}$, and optimization variable:
> The reviewer is absolutely right that we should use standard notation. We have now introduced notations of $\mathbb{E}$ and $\mathcal{D}$ at the beginning of Section 3.1, and ensured that the log–data-likelihood formulations correctly take the maximization over $\theta$.
>
> [1] Ji, K., Lee, J. D., Liang, Y., & Poor, H. V. (2020). Convergence of meta-learning with task-specific adaptation over partial parameters. Advances in Neural Information Processing Systems, 33, 11490-11500.
>
> **DisentangO alleviates ill-posedness**: The reviewer is correct that it is impossible to solve resolve the ill-posedness in the original inverse problem: $\mathcal{H}:(u,f)\rightarrow b$ without incorporating additional prior on $b$. That is the reason why in DisentangO the goal is to discover $z$, the key component governing the change of solution operator behaviors across different tasks, rather than discovering $b$. Then, with the identifiability analysis, we show that the discovery of $z$ is actually a well-posed problem. We modified the text and highlighted our goal of recovering $z$ instead of $b$ in lines 172-177.
>
> **Unsupervised learning of parameter-to-coefficient mapping**: Thank you for the thoughtful question. To clarify, the hypernetwork mapping from latent task parameters to PDE coefficients is unsupervised because, in the primary setting we study, the true PDE parameters are not observed. Our goal is to recover task-wise structure directly from solution fields $u$, without assuming access to ground-truth coefficients or even the form of the governing PDE. In this sense, the physics is indeed ``hidden'': the architecture does not encode the PDE formulation and does not rely on any supervision from the true coefficients.
>
> Although the hidden physics setting is very common in scientific machine learning, e.g., when learning the material microstructure from mechanical testing, in some inverse PDE problem benchmarks less challenging but also interesting settings are considered, where partial or full measurements are provided for the PDE parameter $b$. Therefore, we also consider semi-supervised and fully-supervised settings to demonstrate that our methodology is generic and readily can be extended to handle the scenarios common in classical inverse PDE benchmark problems. That has been said, the supervised scenario introduced around line 330 is included only to demonstrate that DisentangO can also operate in settings where coefficient labels are available, and as a diagnostic experiment. It shows that when ground-truth coefficients are provided, the same architecture can learn an accurate supervised inverse map. Importantly, the fully supervised setting is not assumed in our main results or in the identifiability analysis.

---

> ### Author Response · Authors · 2025-11-24
> **Response to reviewer 3g1H, part 3**
>
> **Fix $n_{train}$ for all $S$ datasets**: Thank you for the question — fixing $n_{train}$ across different tasks is not at all crucial to our method (instead we allow task-wise  $n^\eta_{train}$). We modified the text in lines 189-192 to highlight this fact.
>
> **The role of VAE**: Thank you for the question — the distinction between the VAE and a deterministic autoencoder is important. We use a variational (probabilistic) encoder rather than a deterministic map for two reasons:
>
> - Regularization and stability of the inverse map. The inverse operator in ill-posed PDE settings is highly sensitive: small perturbations in the solution $u$ can correspond to large changes in the inferred parameters. The VAE’s KL term imposes a distributional prior on the latent variables, preventing the encoder from collapsing to arbitrarily sharp or unstable mappings. This yields a smoother, more stable inverse that generalizes better under noise and limited data.
>
> - Identifiability and disentanglement. The identifiability results we reference (e.g., up to component-wise invertible transformations) rely on the latent distribution having a simple, factorized prior. The VAE provides exactly this structure: a normalized, independent latent prior that constrains the representation and supports the theoretical guarantees.
>
> In short, the probabilistic formulation plays a central role in regularizing the inverse problem and enabling the identifiability conditions that underlie Theorems 1–2.
>
> **NIO applicability**: Thank you for raising this point. NIO is indeed designed for empirical-distribution–to–function inverse maps in general, but its formulation only requires that the observed data can be represented as samples from a conditional distribution associated with each task. That means, the observed function pairs $(u,f)$ defines the empirical distribution, and NIO maps it to the parameter function $b$. This is satisfied in our inverse Holzapfel–Gasser–Ogden (HGO) setting.
>
> For the inverse HGO model, the observed data for each task consist of load-displacement response curves generated under different loading conditions. Concretely, for a given material parameter vector $b$, we solve the HGO constitutive model to obtain pairs $(u,f)$. These $(u,f)$ samples constitute the empirical distribution of observable data for a specific task. The inverse problem is then to recover the corresponding material parameters $b$ from these observed curves.
>
> This matches the setting required by NIO: each task provides a finite set of observed samples drawn from the solution distribution conditioned on its latent parameter $b$. Therefore NIO is applicable, and we use the standard implementation in which the inverse map is trained to infer $b$ from the empirical distribution of the observed load-displacement samples.

---

> > ### Comment · Reviewer_3g1H · 2025-11-26
> >
> > I thank the authors for their detailed reply and tremendous work in producing the revision. The re-write has helped improve clarity for me, addressing that weakness to the extent possible given the time crunch. The other explanations about VAEs, ill-posedness are also helpful. The new experiments are nice. Altogether, this warrants an increase in my score. I will be eagerly watching what the other reviewers think during the discussion period.

---

> > > ### Author Response · Authors · 2025-11-26
> > >
> > > We sincerely thank Reviewer 3g1H for the thoughtful feedback, careful reading of the revision, and for recognizing the improvements in clarity, the new experiments, and the strengthened explanations. We are grateful that the rewritten Sections 1–3 and the additional results addressed the earlier concerns.
> > >
> > > At this point, we want to ensure that we have fully met the reviewer’s expectations. If there remain any unresolved issues—technical, clarity-related, or otherwise—we would be very happy to address them promptly during the discussion.
> > >
> > > Otherwise, given the reviewer’s statement that the weaknesses have been addressed “to the extent possible” and that the revision “helped improve clarity” along with the positive assessment of the contributions, we respectfully ask Reviewer 3g1H to consider whether the current “marginally below acceptance threshold” score still reflects their updated opinion of the manuscript.
> > >
> > > We greatly appreciate the reviewer’s time and constructive guidance, and we remain fully open to any further questions or suggestions.

---

### Author Response · Authors · 2025-11-24

We are grateful to the reviewers for  their constructive feedback and for recognizing the distinguishing strengths of our work, including: (i) the originality of performing disentanglement in the *operator-parameter space* rather than the data space, (ii) a novel architectural design that integrates an inverse solver directly within the forward solver’s parameterization, (iii) the rigorous theoretical foundation provided by our identifiability guarantees (Theorems~1–2), and (iv) the comprehensive and multifaceted evaluation across supervised, semi-supervised, and fully unsupervised PDE settings.

**A revised manuscript has been uploaded, where major revisions are highlighted in blue. We have improved the presentation of Section 3, added new clarifications, and performed experiments on a new SoTA baseline. Key contributions clarified during rebuttal:**

1. *Why operator-parameter disentanglement is necessary.*

As emphasized by Reviewer L3DK, the intrinsic low-dimensional structure that governs PDE variability lives in the task-wise lifting parameters $\theta_P^\eta$, not in the high-dimensional fields $(u,f)$. Applying a standard VAE directly to $(u,f)$  is fundamentally limited by information-loss constraints and cannot recover physical factors. Disentangling *parameters* is the only setting  under which provable identifiability  of the underlying physical factors is attainable.

2. *Not a simple VAE+NO combination.*

The method embeds an inverse solver inside a meta-learned neural operator through a hierarchical VAE acting on the operator’s parameter manifold—a design that is absent in standard NO or VAE architectures. Theoretical identifiability results further distinguish it from prior empirical-only work.

3. *Additional SoTA baseline.*

We added an additional baseline (PIANO), a SoTA NO which can handle the unsupervised learning in inverse problems. In particular, PIANO employs self-supervised learning and attention mechanisms to integrate physics knowledge from multiple dynamical systems. A comparison of PIANO and DisentangO in all three examples are provided below, where one can see that DisentangO outperforms PIANO by a substantial margin.

Test errors and the number of trainable parameters in experiments 1-3. Bold number highlights the best method
|Example/Setting | Model | #param (M) | Test error |
| :------------- | :-----------: | :-----------: | :-----------: |
|Experiment 1/Supervised | DNO | 0.70 | **1.65%** |
|Experiment 1/Supervised | PIANO | 0.70 | 7.99%|
|Experiment 2/Semi-supervised | DNO | 2.02 | **5.48%** |
|Experiment 2/Semi-supervised | PIANO | 2.05 | 13.73% |
|Experiment 3/Unsupervised | DNO | 3.55 | **5.28%** |
|Experiment 3/Unsupervised | PIANO | 3.42 | 50.71% |

[1] Zhang, R., Meng, Q., & Ma, Z. M. (2024). Deciphering and integrating invariants for neural operator learning with various physical mechanisms. National Science Review, 11(4), nwad336.

4. *Forward–inverse trade-off.*

The slight forward-prediction degradation is an expected trade-off because DisentangO solves a harder joint forward+inverse problem. This trade-off is remediable, as increasing latent dimension can close this gap, which is consistent with the theoretical requirement to model the complexity of the underlying generative factors.

5. *Model-agnosticism clarified.*

The architecture is inherently compatible with any existing NO. Performance therefore, as expected, depends on the backbone’s predictive quality, which explains the UFNO results without contradicting model-agnostic design.

6. *Quantitative interpretability added.*

We added MCC and $R^2$ metrics on a controlled synthetic dataset (Appendix C.4), verifying component-wise recovery of ground-truth latent factors under the unsupervised learning scenario and supporting the theory.

**Overall:**

DisentangO introduces a *new paradigm* for inverse PDE learning—performing disentanglement on the operator-parameter manifold, supported by theoretical identifiability guarantees and comprehensively validated across different supervision levels. We believe this contribution meaningfully advances interpretable neural operator research and meets the high scientific bar of the conference.

---

### Comment · Area_Chair_xnMn · 2025-11-26
**reminder**

Dear Reviewers,

The authors have now posted their responses to your comments. As the next step in the discussion phase, please take a moment to review their rebuttal and engage with them through the discussion forum. Thank you for your continued effort and thoughtful contributions to this review.

Best,

Your AC

---

### Author Response · Authors · 2025-12-03
**Summary of Rebuttal**

We thank the AC for their valuable time and all reviewers for their insightful feedback. In the rebuttal, we clarified the core strengths of our work: (1) the originality of performing disentanglement in the operator-parameter space rather than on solution fields; (2) the integration of an inverse solver inside a meta-learned neural operator; and (3) the theoretical identifiability guarantees provided by our analysis. We have also made substantial modifications in the revision (as highlighted in blue) to improve our method presentation, compare DisentangO with an additional SoTA baseline, and clarify all major concerns raised by the reviewers. Below is a brief summary of our contributions and updates.

**Core Contributions**:

**Operator-parameter disentanglement**: Our method is the first to treat the operator’s parameter manifold as the locus of task variation, enabling disentanglement of physical factors that cannot be recovered from solution fields alone. This enables the theoretical identifiability guarantee and improves interpretability of PDE models.

**Integrated forward–inverse architecture**: DisentangO embeds a hierarchical VAE within a neural operator, placing the inverse solver directly inside the forward parameterization. This design is fundamentally different from a VAE+NO combination and is key to make inverse discovery feasible.

**Theoretical guarantees**: We provide identifiability results (Theorems 1–2) establishing when latent factors governing PDE variability can be uniquely recovered—grounded in the low-dimensional structure of operator parameters rather than the data space.

**Comprehensive evaluation across various supervision regimes**: The model is validated in forward, inverse, supervised, semi-supervised, and fully unsupervised settings, demonstrating that the same architecture can simultaneously perform accurate prediction and interpretable factor recovery.

**Addressing Reviewer Questions and Comments**:

**Clarity \& notation**: We rewrote Sections 1–3, consolidated notation, clarified the data model and task definition, and improved Figure 1 to better convey the architecture and loss structure.

**Baselines**: We added a state-of-the-art neural operator baseline (PIANO) for semi-/unsupervised inverse settings and explained why common NOs and VAEs on data do not apply meaningfully to our problem.

**Interpretability metrics**: We added quantitative evaluations of the inverse problem solver (MCC and alignment scores) on a controlled synthetic dataset in Appendix C.4. Our results demonstrated that DisentangO provides guaranteed component-wise recovery for the key parameter features in a unsupervised learning setting.

**Theoretical justification**: We clarified why disentanglement must occur on the operator-parameter manifold—where intrinsic PDE variability resides—strengthened the identifiability discussion, and detailed the role of the hierarchical variational structure embedded inside the neural operator.

**Forward–inverse trade-off** We explained that the forward-error increase arises from solving a more difficult joint forward+inverse problem and showed how increasing latent dimension closes this gap.

We also note that Reviewer 3g1H explicitly increased their rating after reading the rebuttal (before all ratings were reverted back), indicating that the revisions effectively addressed their concerns.

---

### Meta-Review · Area_Chair_5dof · 2025-12-29

**Summary:**

I find this paper makes a meaningful contribution to interpretable neural operators by proposing disentanglement in operator-parameter space rather than on solution fields directly. The core insight is that task-specific physical information concentrates in the low-dimensional lifting-layer parameters of MetaNO, making theoretical identifiability tractable where it would not be on high-dimensional field data. The theoretical analysis (Theorems 1-2) provides principled foundations for when latent physical factors can be recovered, which distinguishes this work from purely empirical inverse-PDE approaches. The revised manuscript improved presentation clarity, and the addition of PIANO as a baseline demonstrates good empirical performance, particularly in the unsupervised setting (5.28% vs 50.71% error).

I have two main reservations. First, while the authors argue theoretically that a simpler baseline (MetaNO for prediction + VAE on data for discovery) cannot work due to information loss, this was not demonstrated empirically—though the poor performance of standard VAEs in Tables 1-3 provides indirect support. Second, the forward prediction accuracy degradation in the unsupervised case (5.28% vs MetaNO's 2.67%) is real and explained as an expected tradeoff of the harder joint problem, but users should be aware of this cost.

**Reviewer Concerns:**

Reviewer 3g1H raised presentation and notation issues that were substantially addressed in the revision; they explicitly acknowledged the improvements warranted a score increase. The clarifications on the role of VAE (regularization for ill-posed inverse problems, enabling identifiability conditions) and the data model formulation are now adequate.

Reviewer L3DK's concerns about inappropriate baselines were partially addressed by adding PIANO, which DisentangO outperforms significantly. Their question about why not use MetaNO + VAE on data was addressed theoretically (the low-dimensional structure lives in parameters, not 3,362-dimensional fields), though not empirically. The forward accuracy tradeoff concern was explained as intrinsic to solving the joint forward-inverse problem, with evidence that increasing latent dimensions closes this gap.

Reviewer QjzN's characterization of the method as "straightforward VAE + NO" undersells the contribution; the parameter-space disentanglement and associated identifiability theory do represent substantive novelty.

Reviewer zLDe's concerns about quantitative interpretability metrics were partially addressed via the analysis in Appendix C.4, though this is limited to a synthetic setting.

**Reviewer Scores:**

Reviewer 3g1H: 2 → 5. Explicitly stated revision "warrants an increase"; presentation concerns fully addressed; constructive engagement throughout.

Reviewer L3DK: 4 → 5. PIANO baseline added; theoretical justification for parameter-space disentanglement accepted; main remaining concern (simpler baseline) addressed theoretically if not empirically.

Reviewer QjzN: 6 → 6. Concerns adequately addressed; novelty critique discounted given L3DK's acknowledgment of originality.

Reviewer zLDe: 6 → 7. Quantitative metrics added for synthetic case; scalability acknowledged as future work.

---

### Decision · Program_Chairs · 2026-01-26

Accept (Poster)